# Old age variably impacts chimpanzee engagement and efficiency in stone tool use

Elliot Howard-Spink[1,2,3]*, Tetsuro Matsuzawa[4,5], Susana Carvalho[6,7,8], Catherine Hobaiter[9], Katarina Almeida-Warren[7,10], Thibaud Gruber[11†], Dora Biro[12†]

[1]Department of Biology, University of Oxford, Oxford, United Kingdom; [2]Development and Evolution of Cognition Group, Max Planck Institute of Animal Behavior, Konstanz, Germany; [3]School of Biological and Behavioural Sciences, Queen Mary University of London, London, United Kingdom; [4]Department of Pedagogy, Chubu Gakuin University, Gifu, Japan; [5]College of Life Science, Northwest University, Xi'an, China; [6]Department of Science, Gorongosa National Park, Sofala, Mozambique; [7]Interdisciplinary Center for Archaeology and Evolution of Human Behaviour (ICArEHB), Universidade do Algarve, Faro, Portugal; [8]CIBIO, Centro de Investigação em Biodiversidade e Recursos Genéticos, Vairão, Portugal; [9]Wild Minds Lab, School of Psychology and Neuroscience, University of St Andrews, St Andrews, United Kingdom; [10]School of Anthropology and Museum Ethnography, University of Oxford, Oxford, United Kingdom; [11]Faculty of Psychology and Educational Sciences, and Swiss Center for Affective Sciences, University of Geneva, Geneva, Switzerland; [12]Department of Brain and Cognitive Sciences, University of Rochester, Rochester, United States

*For correspondence: elliot.howardspink@outlook.com

†These authors contributed equally to this work

Competing interest: The authors declare that no competing interests exist.

## eLife Assessment

This **valuable** study provides a novel framework for leveraging longitudinal field observations to examine the effects of aging on stone tool use behaviour in wild chimpanzees. The methods and results are robust providing **solid** evidence of the effects of old age on nut cracking behaviour at this field site. Despite the low sample size of five individuals, this study is of broad interest to ethologists, primatologists, archaeologists, and psychologists.

**Abstract** We know vanishingly little about how long-lived apes experience senescence in the wild, particularly with respect to their foraging behaviors. Chimpanzees use tools during foraging, and given the cognitive and physical challenges presented by tool use, tool-use behaviors are potentially at a heightened risk of senescence, though this has never been investigated in wild individuals. Accordingly, we sampled data from a longitudinal video archive that contained footage of wild chimpanzees using stone hammers and anvils to crack hard-shelled nuts (*nut cracking*) at an 'outdoor laboratory' over a 17-year period (with focal chimpanzees aging from approximately 39–44 to 56–61 years across this period). Over time, elderly chimpanzees began attending experimental nut-cracking sites less frequently than younger individuals. Several elderly chimpanzees exhibited reductions in efficiency across multiple stages of nut cracking, including taking longer to both select stone tools prior to use and use tools to crack open nuts and consume the associated pieces of kernel. Two chimpanzees began using less streamlined behavioral sequences to crack nuts, including a greater number of actions (such as more numerous hammer strikes). Notably, we report interindividual variability in the extent to which elderly chimpanzees' tool-use behaviors changed during our

sample period – ranging from small to profound reductions in engagement and efficiency – as well as differences in the specific aspects of nut cracking that changed for each individual. We discuss the possible causes of these changes – and recommendations for future research – with reference to literature surrounding the senescence of captive and wild primates.

## Introduction

Senescence can be understood as a reduction in the efficacy of biological systems with increasing age. In recent decades, the senescence of non-human primates (henceforth primates) has been a point of considerable research interest (*Colman, 2018*; *Edler et al., 2021*; *Gilleard, 2023*). In part, this is due to the fact that primates offer uniquely translatable models for understanding the nature and evolution of human senescence, given their high genetic, physiological, and behavioral similarities (*Colman, 2018*; *Bizon and Woods, 2009*; *Frye et al., 2022*; *Lane, 2000*). Accordingly, many parallels between the senescent processes of humans and primates have been uncovered, including in their cognition (*Comrie et al., 2018*; *Gray and Barnes, 2019*; *Herndon and Lacreuse, 2002*; *Lacreuse et al., 2014*; *Lacreuse et al., 2018*; *Lacreuse et al., 2020*) physiology (*Colman, 2018*; *Colman and Binkley, 2002*; *Didier et al., 2016*; *Lowenstine et al., 2016*), and their resultant behaviors (*Havercamp et al., 2021*; *Hozer et al., 2019*; *Neal Webb et al., 2019*; *Tarou et al., 2002*; *Zhdanova et al., 2011*). However, thus far, the majority of these studies have focused on captive populations of short-living species and exploited existing variation in age among individuals to infer the effects of senescence through cross-sectional analyses (*Colman, 2018*; *Gray and Barnes, 2019*; *Rothwell et al., 2021*; *Tardif and Ross, 2021*). Comparatively, few studies have investigated how senescence influences the behaviors of primates in the wild (with the exception of a handful of examples *Fujisawa et al., 2010*; *Newman et al., 2023*; *Siracusa et al., 2022b*; *Siracusa et al., 2024*; *Thompson et al., 2020*), and even fewer employ longitudinal data sampling to understand how senescence leads to changes in the behaviors of specific individuals over time. These studies of wild individuals are highly valuable as they shed light on how aging affects the day-to-day behaviors that primates rely on for survival – conclusions which cannot be easily predicted from studies from captivity, given that patterns of physiological senescence can differ significantly between captive and naturalistic environments (*Cole et al., 2024*; *Campos et al., 2024*). Further research is therefore required to understand how aging influences the lives of wild primates. This is particularly true for long-living great ape species, which are underrepresented in the existing literature, yet their phylogenetic proximity to humans means that they likely offer the most suitable models for understanding the evolution of the human aging process.

Of the day-to-day behaviors of great apes, the combined physical and cognitive demands of tool use make it likely to exhibit particular changes with progressive old age. The tool-use behaviors of great apes are highly challenging, requiring tool users to draw upon a wide array of high-level cognitive abilities, including planning and the flexible assembly of actions into goal-directed behaviors (*Byrne et al., 2013*; *Call, 2013*; *Carvalho et al., 2008*; *Howard-Spink et al., 2024*); an understanding of causal relationships between objects (*Hayashi, 2015*; *Matsuzawa, 1996*; *Sanz and Morgan, 2010*); as well as knowledge of the physical properties of objects and how to exploit these properties to use tools successfully (*Carvalho et al., 2008*; *Boesch and Boesch, 1983*; *Bril et al., 2009*; *Sirianni et al., 2015*). Many of these cognitive abilities – including the more specific subcomponents of these abilities, such as motor coordination (*Lacreuse et al., 2014*), working memory (*Glavis-Bloom et al., 2022*), executive functioning (*Lacreuse et al., 2020*), and cognitive flexibility (*Lacreuse et al., 2018*) – have been identified as at risk of senescence in captive living primates; however, never in tool-use behaviors themselves. Moreover, physiological changes including reduced bone mass (*Colman and Binkley, 2002*; *Lowenstine et al., 2016*; *Nichols and Zihlman, 2002*), muscle wasting (sarcopenia; *Lowenstine et al., 2016*; *Cann, 2015*; *Morbeck et al., 2002*), the development of arthritis (*Lowenstine et al., 2016*), and reduced visual acuity (*Fujisawa et al., 2010*) have been identified in a number of primate species, including within the great apes, which all influence individual strength, dexterity, and accuracy of movement. These findings suggest that tool use could be a domain of great-ape behavior that is at specific risk of senescence; however, it has not yet been possible to address this question due to an absence of long-term data on great ape tool-use behaviors that also include instances of tool use performed by elderly individuals.

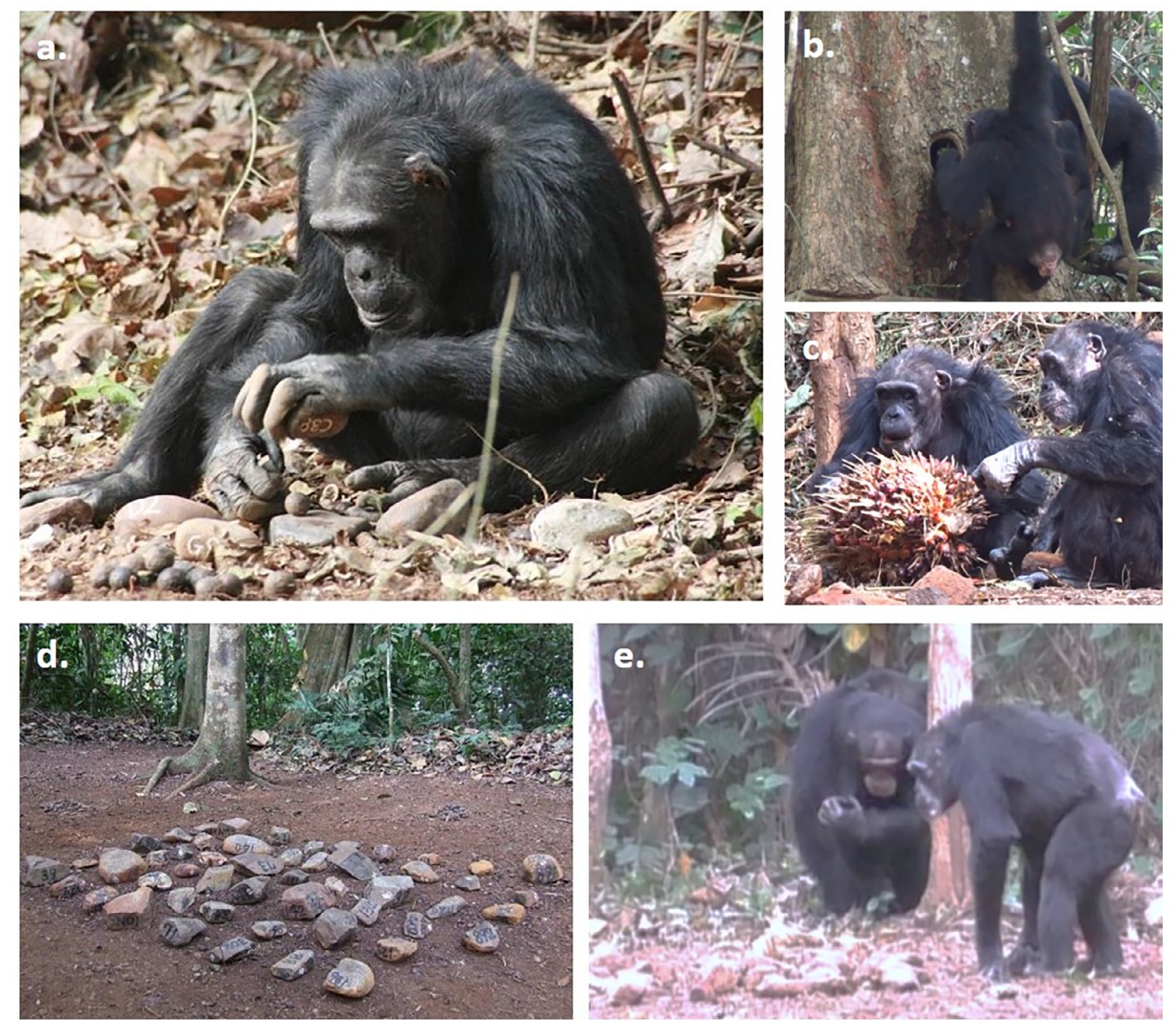

**Figure 1.** The experimental set-up and behaviors at the outdoor laboratory. (**a**) Yo, an elderly female (approximately 51 years old), has placed a coula nut (*Coula edulis*) on an anvil stone and is preparing to strike with the hammer stone. (**b**) An adult female (14 years old) using a leaf tool to drink water from the water point. (**c**) Yo and Velu (left to right, both approximately 56–58 years old), eating oil-palm fruits from an available raceme. (**d**) The central stone-tool matrix with numbered stones. (**e**) Yo (right; approximately 49 years old) selecting tools from the stone matrix.

To address this gap in the literature, we used a longitudinal archive of video footage of wild, West African chimpanzees (*Pan troglodytes verus*) using stone tools to crack hard-shelled nuts as part of a decades-long field experiment in a clearing in the Bossou forest, Guinea (*Matsuzawa et al., 2011*; *Sugiyama, 1981*) ('the outdoor laboratory'; see Methods and *Figure 1* for information about the experimental set up at the outdoor laboratory). Nut cracking is one of the most sophisticated forms of tool use observed in the animal kingdom, and is habitually performed by a number of primate species in the wild including several communities of chimpanzees (*Pan troglodytes*; *Boesch and Boesch, 1983*; *Carvalho and McGrew, 2012*; *Matsuzawa, 1994*), capuchin monkeys (*Cebus* and *Sapajus* spp.; *Falótico et al., 2019*; *Fragaszy et al., 2023*; *Falótico et al., 2024*; *Goldsborough et al., 2024*; *Goldsborough et al., 2025*), and long-tailed macaques (*Macaca fascicularis*; *Luncz et al., 2019*; *Proffitt et al., 2018*). During nut cracking, a nut is placed on a hard substrate for the chimpanzee population used in this study, this substrate is a portable anvil stone, although at other sites this may be a root or rocky outcrop (*Boesch and Boesch, 1983*; *Sirianni et al., 2015*), before being struck repeatedly with a hammer stone until the nut is cracked open and the edible kernel inside extracted and consumed (see *Figure 1a*). Nut cracking is learnt by chimpanzees at Bossou by the age of 7 years

(often between 3.5 and 5 years of age; *Biro et al., 2006*; *Inoue-Nakamura and Matsuzawa, 1997*), and chimpanzees continue to perform this behavior throughout adulthood (*Matsuzawa et al., 2011*). This extended engagement in nut cracking permits characterization of how individual chimpanzees perform this behavior across their lives (*Berdugo et al., 2025*), including during the oldest years of their lifespan. Nut cracking features all of the key elements of tool-use behaviors which we predict will make them likely to senesce with old age, including the need to construct complex object associations (*Hayashi, 2015*; *Matsuzawa, 1991*; *Matsuzawa, 1996*) to organize and address multiple goals using an extended behavioral sequence (*Carvalho et al., 2008*; *Howard-Spink et al., 2024*) to select tool objects based on perceived properties (*Carvalho et al., 2008*; *Boesch and Boesch, 1983*; *Sirianni et al., 2015*), and to combine objects with sufficient dexterity, precision and force and to crack nuts without knocking nuts off the anvil, or smashing interior kernels (*Bril et al., 2009*).

To determine whether – and if so, how – old age influenced the nut-cracking behaviors of wild chimpanzees, we sampled video footage collected at the outdoor laboratory in five field seasons spanning a 17-year timeframe between 1999 and 2016. During this timeframe, we collected longitudinal data on five chimpanzees as they aged from approximately 39–44 years old (end of mid-life) to 56–61 years old (ages which are close to the maximum for wild chimpanzee lifespans; *Wood et al., 2023*). Across sampled field seasons, we estimated how readily chimpanzees engaged in nut cracking behaviors using two key metrics: the frequency with which chimpanzees visited outdoor laboratory sites during each field season (compared to a control group of younger individuals between the ages of 8 and 30 years of age), and when present, the proportion of time chimpanzees spent engaging with nuts and stone tools compared to other key behaviors (such as drinking water from an available water point using leaf tools (*Sousa et al., 2009*) and eating oil-palm fruits that are provisioned alongside nuts; see *Figure 1b and c*). We also measured how quickly chimpanzees selected stone tools from a central matrix of over 50 stones (a generalized metric of efficiency during tool selection; see *Figure 1d and e*), as well as how efficiently chimpanzees cracked open nuts using stone tools and consumed the associated kernel.

Overall, with increasingly old age (measured by progressive field seasons), elderly chimpanzees were significantly less likely to visit experimental nut-cracking sites. In comparison, younger individuals (i.e. younger adults and older immatures who had learnt how to crack nuts) exhibited no change over successive field seasons, suggesting that this reduced attendance at experimental nut cracking sites was confined to an elderly, aging cohort of chimpanzees. When present at the outdoor laboratory, two of our five elderly individuals spent substantially less time engaging in nut cracking at older ages, as compared to their behaviors in previous field seasons. Moreover, several elderly chimpanzees took more time to select stone tools in later field seasons than in earlier field seasons, and were also less efficient at using stone tools to crack open oil-palm nuts. Importantly, we detected considerable interindividual variability in the effects of aging on how readily elderly chimpanzees engaged with nuts and stone tools, how quickly they selected tools, and the efficiency with which elderly chimpanzees used tools. Some individuals demonstrated little-to-no change in engagement and efficiency with aging, whereas others experienced significant changes across different aspects of their nut-cracking behavior. We discuss our findings in the context of existing literature for primate aging and hypothesize which factors underpin observed changes in behavior, including cognitive, physiological, and motivational causes, and generate a number of suggested avenues for further study. We also discuss how aging can likely compound existing interindividual differences in the proficiency of socially-learnt technical skills in non-human animals, and discuss how tool use could potentially present a suitable indicator for identifying individuals who are experiencing the effects of profound senescence in the wild.

## Results
### Sampled data

We sampled five timepoints separated by intervals of 3–5 years (field seasons 1999–2000, 2004–2005, 2008–2009, 2011–2012, and 2016–2017; with each field season referred to hereafter by its initial year). The exact age at which chimpanzees may be considered 'old' is a point of continued debate. However, chimpanzees are generally considered to begin entering old age at approximately 40 years old (*Morbeck et al., 2002*; *Campos et al., 2022*; *Finch and Austad, 2015*), after which survivorship

begins to decrease more rapidly than earlier in adulthood (*Wood et al., 2023*). We therefore confined our analysis to individuals who were at least 30 years of age in at least three of the five sampled field seasons. This sample allowed us to collect data longitudinally, beginning with ages where chimpanzees are in the prime of adulthood and spanning into old age. Under these criteria, we were able to include four old-age females (Fana, Jire, Velu, and Yo) and one old-age male (Tua) in our analyses. All five of these individuals were present at Bossou when the long-term research project was established in 1976; as a consequence, their birth years are estimates based on physical growth characteristics at the time of first observation (see *Table 1*). Given that these estimated birth years range from 1956 to 1961, these individuals are estimated to be between 39 and 44 years old at the start of our sampling window (1999), and therefore are entering the window of 'old age'. By 2016, estimated ages ranged from 56 to 61 years old, reflecting an age which is near the maximum for wild chimpanzees. All four females were present at Bossou throughout the entire sampled time-frame for our analysis; however, Tua, the only adult male, disappeared in September 2013 (presumed dead). Therefore, data for Tua spans a shorter time frame between the 1999 and 2011 field seasons. As all five focal individuals' ages are within five years of each other (and therefore similar estimates), we use the progression through field seasons as a proxy for increasing age and treat all individuals as similarly 'old' at each field season.

A summary of sampling effort for each chimpanzee in each field season can be found in *Table 1*, including their estimated ages in each field season (and subsequent year of death), the number of encounters sampled for behavior coding for each individual (between 10 and 1 per individual per field season; median = 10), the total duration each individual was observed during the field season (mean = 280.6 min per individual per field season; SD = 83.3 min), and the number of nuts that they were observed cracking (between 141 and 5 nuts per year; median across individuals and field seasons = 80).

## Attendance at the outdoor laboratory

We modeled the rate at which focal old-aged chimpanzees attended the outdoor laboratory over progressive field seasons, whilst controlling for the total length of each field season. We compared this relationship with attendance data from younger individuals (between the ages of 8 and 30 years; see *Figure 2a*). This younger cohort acted as a baseline control for changes in attendance rate at the population level that are unlikely to be due to senescence. In each sampled field season, the number of individuals over the age of 30 (older cohort) varied between five and six individuals, whereas the number of chimpanzees between the ages of 8 and 30 years (younger cohort) varied between five and two individuals.

In 1999, chimpanzees in the older cohort had a marginally lower attendance rate than the younger cohort (encounters/day: older cohort = 0.68; younger cohort = 0.74). However, by 2016, this difference was much larger, with older chimpanzees being less likely to visit the outdoor laboratory compared to younger individuals (encounters/day in 2016: older cohort = 0.32; younger cohort = 0.53; see *Figure 2b* for data from each elderly individual). Correspondingly, a model of attendance rate over successive field seasons (including data across all field seasons) identified a significantly greater decline in attendance rate for the older cohort across successive field seasons as compared to the younger cohort (interaction effect of field season and age-cohort: z=–2.285; p=0.022; see *Appendix 2—table 1* for full model output). Conversely, we found no evidence of a decline in attendance rate for chimpanzees in our younger cohort over successive field seasons (z=–1.572; p=0.11). We confirmed the importance of this interaction effect by comparing the fit of the full model with a model lacking the interaction effect between age-cohort and field season (therefore assuming an identical relationship across field seasons for both cohorts) and a null model (which assumed no effect of field season). AIC comparison confirmed that the model which included an interaction effect offered the best explanation of the data relative to model complexity ($AIC_{Interaction}$ = 340; $AIC_{Interaction-removed}$=343; $AIC_{Null}$ = 376). All four elderly female chimpanzees exhibited attendance rates that were lower in 2016 than in 1999; however, for Tua (the elderly adult male), attendance rates in his earliest and latest field seasons were approximately equal (1999=0.48 encounters/day; 2011=0.5 encounters/day; see *Figure 2b*).

## Behaviors at the outdoor laboratory

We measured how the proportion of time each elderly individual spent interacting with nuts and stones when present at the outdoor laboratory changed over successive field seasons (see *Figure 2c*).

**Table 1.** Summary of sampled observations for each focal old-age individual.

Total time observed includes all time individuals were present in the first 10 encounters of each field season (Observed Encounters). Dashed lines (-) represent where no data was collected for an individual in a given field season. Males have names in all capitals, whereas females have names in capitals and lower case.

| ID | DOB* | Date of Death (Age) | Age (years) | | | | | Observed Encounters (% including interaction with nuts or stone tools) | | | | | Total Time Observed (minutes) | | | | | Action Sequences Coded (for individual nuts) | | | | |
|---|---|---|---|---|---|---|---|---|---|---|---|---|---|---|---|---|---|---|---|---|---|---|
| | | | 1999 | 2004 | 2008 | 2011 | 2016 | 1999 | 2004 | 2008 | 2011 | 2016 | 1999 | 2004 | 2008 | 2011 | 2016 | 1999 | 2004 | 2008 | 2011 | 2016 |
| Fana | 1956 | 2022 (67) | 44 | 49 | 53 | 56 | 61 | 10 (70%) | 10 (70%) | 10 (80%) | 7 (28.6%) | 10 (40%) | 280.6 | 275.2 | 184.2 | 105.7 | 283.5 | 80 | 77 | 80 | - | 70 |
| Jire | 1958 | Alive as of 2025 (67) | 42 | 47 | 51 | 54 | 59 | 10 (80%) | 10 (80%) | 10 (90%) | 10 (60%) | 10 (80%) | 339.4 | 349.9 | 275.0 | 282.1 | 308.6 | 100 | 91 | 104 | 42 | 82 |
| TUA | 1957 | 2013 (56) | 43 | 48 | 52 | 55 | - | 10 (70%) | 10 (80%) | 10 (80%) | 5 (60%) | - | 174.4 | 212.1 | 204.3 | 116.0 | - | 103 | 96 | 81 | 26 | - |
| Velu | 1959 | 2017 (58) | 41 | 46 | 50 | 53 | 58 | 10 (60%) | 10 (80%) | 7 (85.7%) | 1 (100%) | 7 (42.9%) | 189.4 | 281.8 | 158 | 23.1 | 202.4 | 85 | 99 | 50 | - | 5 |
| Yo | 1961 | 2021 (60) | 39 | 44 | 48 | 51 | 56 | 10 (80%) | 10 (70%) | 10 (90%) | 3 (33%) | 5 (40%) | 200.7 | 304.5 | 363.7 | 211.3 | 282.7 | 66 | 70 | 141 | 20 | 33 |

*Date of birth (DOB) was estimated for these individuals at the start of longitudinal data collection at Bossou (1976). Thus, ages for individuals are estimates. Ages are approximated from January of the estimated year of birth, and the end of the final month of each field season (e.g. for the 1999-2000 field season – abbreviated to 1999 - ages are estimated using 29th February 2000 as the end of the field season).

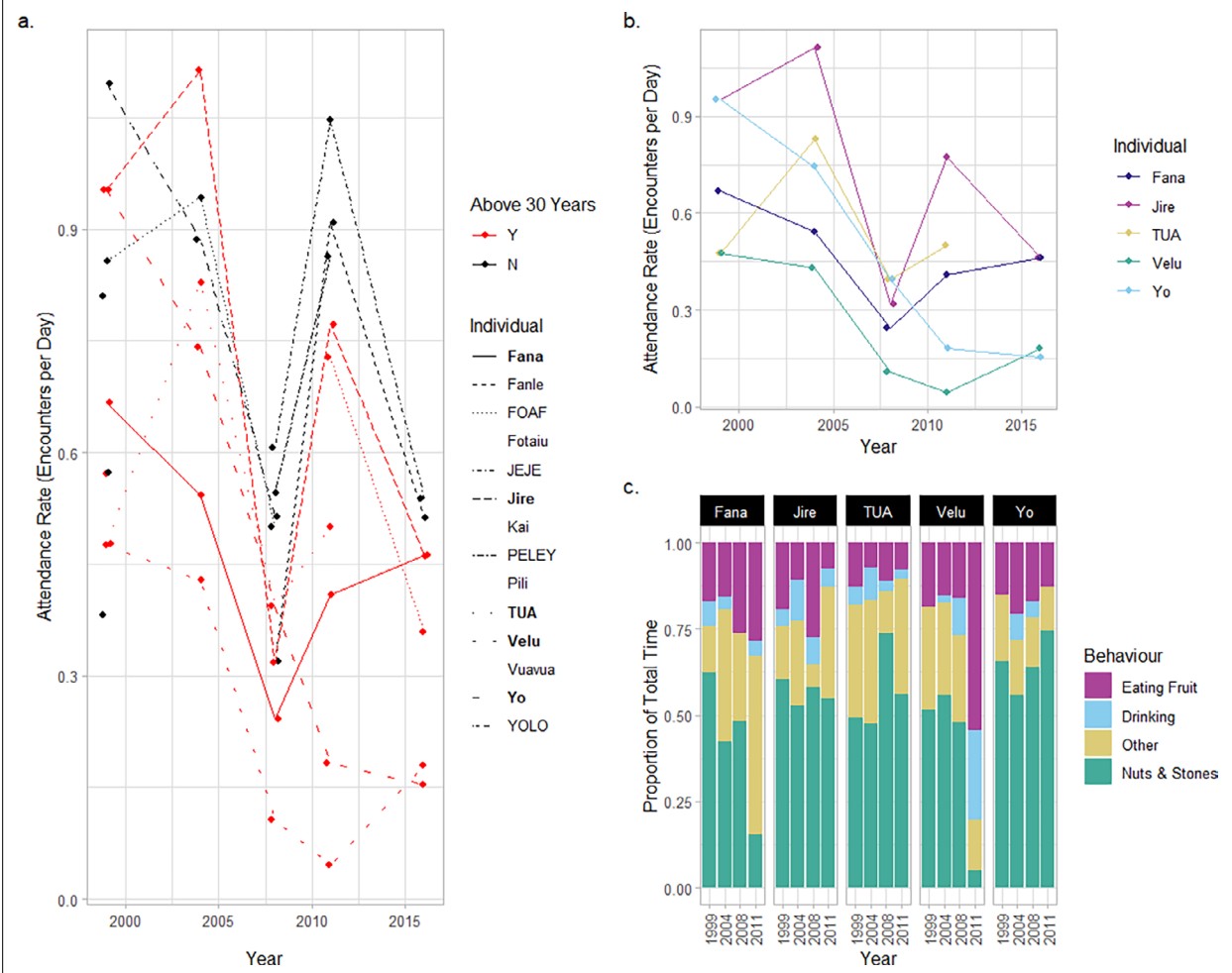

**Figure 2.** Attendance and behavior of old-aged chimpanzees at the outdoor laboratory. (**a**) Attendance rate for individuals at the outdoor laboratory between the 1999 and 2016 field seasons. Red points and lines indicate individuals are over 30 years old; black points and lines indicate individuals younger than 30 years. Lines are drawn for all individuals who attended the outdoor laboratory in two or more field seasons (individuals who were only sampled in one field season have blank spaces in the legend). The names of males are provided in all capitals, and females are provided with capital and lower-case letters. Focal old-aged individuals are indicated in the legend in bold. (**b**) Attendance data for the five focal individuals as they age from 1999 to 2016. (**c**) The proportion of total time individuals spent engaging in four different categories of behavior at the outdoor laboratory between 1999 and 2011 (data collected at the first outdoor laboratory location only).

By 2011, the final year of this analysis (we omitted the use of 2016 for this analysis, see Methods), two old-aged individuals (Fana and Velu) spent substantially less time engaging with nuts and stone tools (% time engaging with nuts and stone tools in each season; Fana: 1999=62.3%; 2011=15.6%; Velu: 1999=51.6%; 2011=5.0%). We did not observe either individual successfully cracking any nuts at the outdoor laboratory in 2011. Most interactions with nuts and stone tools involved scavenging kernels from the ground (produced by other individuals' nut cracking behaviors), or in the case of Velu, a short attempt to crack open a nut, before ceasing the behavior and feeding from oil-palm fruits. Correspondingly, in 2011, Fana spent more time engaging in 'Other' behaviors, such as resting and grooming (+38.3% of total time in 2011 compared with 1999), whereas Velu spent more time eating palm fruits (+35.5%) and drinking water from the water point (+25.8%; note that data for Velu come from a single encounter in 2011, so proportions should be interpreted with caution). Conversely, three individuals spent similar proportions of time engaging with nuts and stone tools when present at the outdoor laboratory (change in % time between 1999 and 2011: Jire = –5.5%; Tua = +6.9%; Yo = +8.9%). For these individuals, the small-scale changes in total time engaging with nuts and stones are more likely to be the product of chance fluctuation in engagement across field seasons.

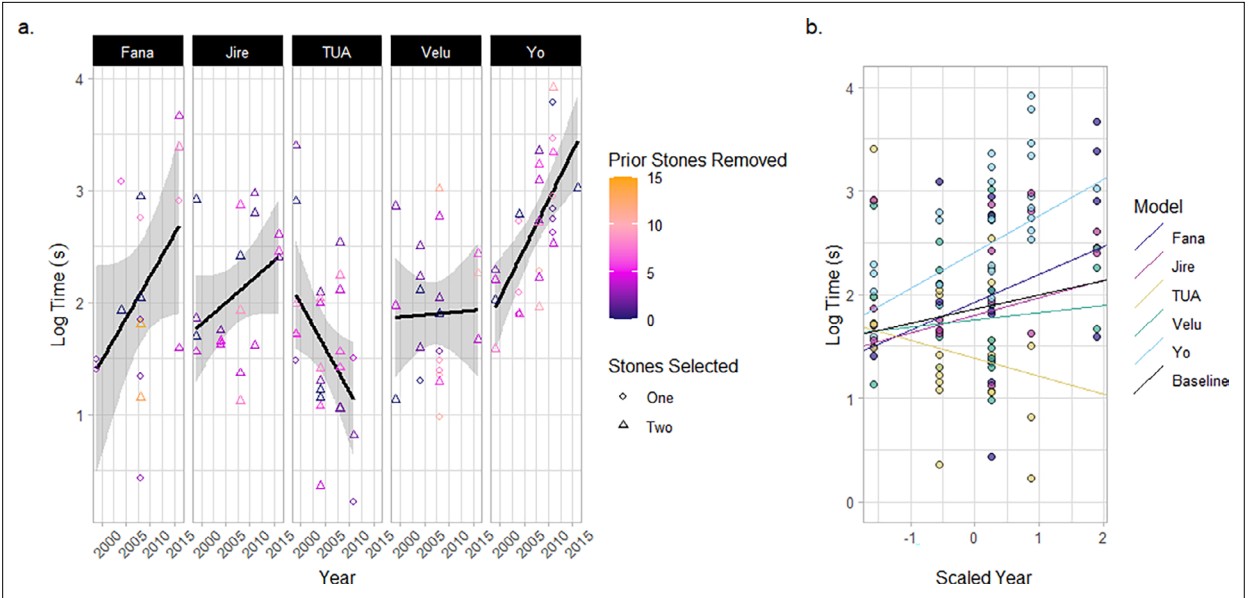

**Figure 3.** Duration of stone-tool selection events. (**a**) Tool-selection duration times for each old-aged individual. Color correlates with the number of stone tools removed from the matrix prior to that particular tool-selection event. Shape indicates the number of tools selected by an individual in a given tool-selection event. The lines and shaded areas represent a smoothed linear relationship describing the data for each individual. (**b**) A mixed effect model describing the duration of stone tool-selection events across a scaled parameter of the year for each field season. Individuals are included in the model as both a random intercept and slope. The plot shows the model's prediction of the relationship between the duration of stone-tool selection and year for each individual, compared with the baseline fixed effect of year.

## Tool-selection time

Chimpanzees select stone tools based on their physical characteristics such as raw material, size, and weight (*Carvalho et al., 2008*; *Boesch and Boesch, 1983*; *Sirianni et al., 2015*; *Braun et al., 2025*). Therefore, stone-tool selection at the central stone matrix represents a complex decision-making task where chimpanzees must evaluate an extensive set of options (over 50 total stones). Between 1999 and 2016, we recorded the duration of 108 stone-tool selection events across 49 encounters for the five focal old-aged individuals (see *Figure 3a*, and *Appendix 2—table 2*). A mixed-effect model with the year of the field season as both a random intercept and random slope for each individual offered the best explanation for the total variation in the time taken to select tools (see *Figure 3b* for a visualization of the random slope model). This model outperformed a null model which assumed no effect of field season ($AIC_{Year-RandomSlope}$=231; $AIC_{Null}$ = 234) as well as a model which included a fixed slope for field seasons – and therefore the same aging effect – across all individuals (where models were refit by restricted maximum likelihood to improve the accuracy of random-effect estimation; $AIC_{Year-Random}$=244; $AIC_{Year-Fixed}$=246; see *Appendix 2—Tables 3–5* for model summaries). The best explanation for our data is therefore one where aging has a significant effect on tool-selection time; however, the effects of aging differ between individuals. The random-slope model identified that Yo underwent the greatest increase in stone-tool selection time across sampled field seasons, followed by Fana and then Jire. Velu exhibited relative consistency in the duration of stone-tool selection across field seasons, and the only individual to show a negative relationship between the duration of stone-tool selection and increasing years was Tua.

## Efficiency processing oil-palm nuts

Across the five focal elderly chimpanzees, we recorded sequences of actions used to crack 1601 individual nuts (1538 oil-palm nuts and 63 coula nuts). As coula nuts were only provided alongside oil-palm nuts in one sampled field season (2011), we filtered data to include only the 1538 oil-palm nuts when modeling the influence of progressive old age on the efficiency of nut cracking and described data on the cracking of coula nuts separately (see Appendix 3).

For three efficiency metrics, models that contained the year of the field season as a fixed effect outperformed null models which assumed no change across years (models were fit using data from

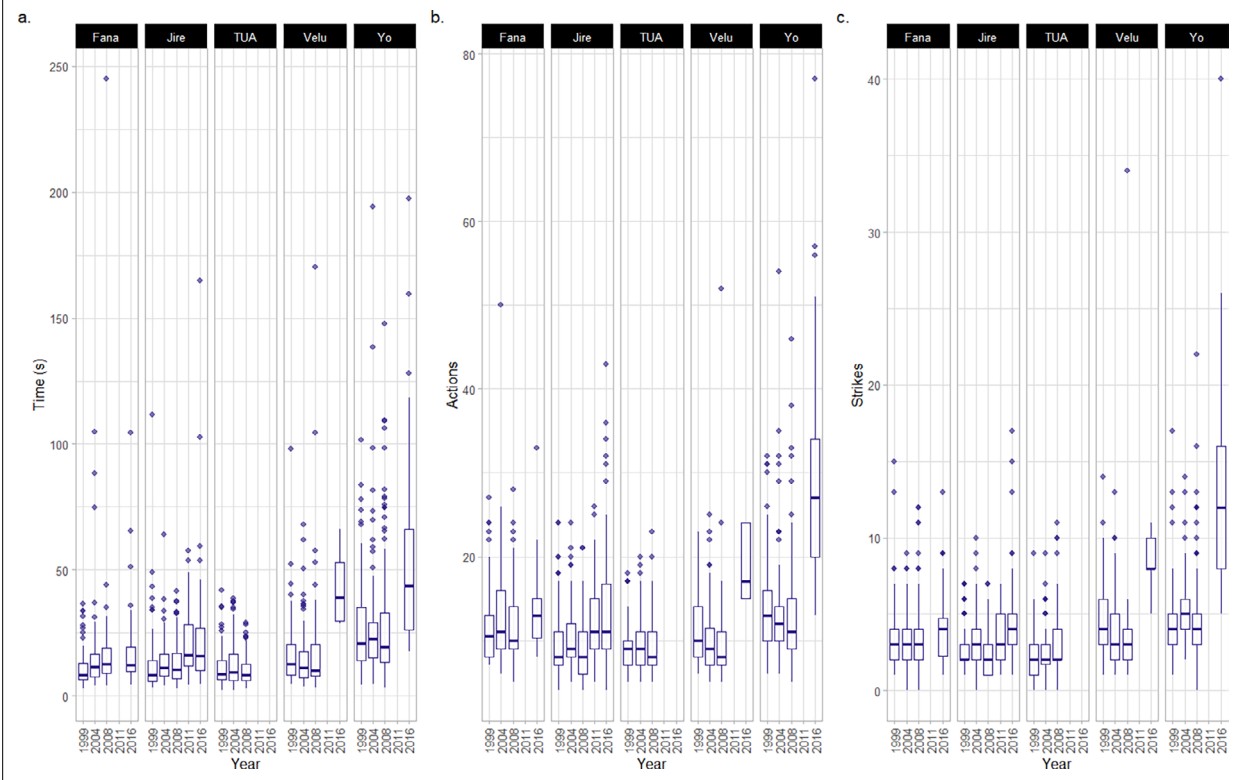

**Figure 4.** Metrics of efficiency for the cracking and processing of oil-palm nuts. This plot only includes metrics in which random slope models outperformed corresponding null models (see *Figure 4—figure supplement 1* for all metrics). Data is summarized using boxplots (central lines in the boxes indicate median values, and upper and lower boundaries of boxes indicate the 25th and 75th percentiles; lines express the range of data, and dots indicate outliers for a given individual in a given year). Sample size available in *Table 1*. All data relate to the cracking and processing of individual oil-palm nuts. These metrics include, for each nut cracked, (**a**) the total time taken, (**b**) the number of discrete actions, and (**c**) the number of hammer strikes.

The online version of this article includes the following figure supplement(s) for figure 4:

**Figure supplement 1.** Summary data on all efficiency metrics for all focal aging chimpanzees.

all field seasons). These metrics included the total time taken to crack and process nuts ($AIC_{Year}$ = 3086, $AIC_{Null}$ = 3092), the total number of actions used during the cracking of each nut ($AIC_{Year}$ = 9452, $AIC_{Null}$ = 9485), and the total number of strikes of the hammer stone performed per nut ($AIC_{Year}$ = 6582, $AIC_{Null}$ = 6617; see *Figure 4—figure supplement 1* for data on all measured efficiency metrics, including those where we found no effect of aging, and *Appendix 2—Tables 6–9* for model summaries for the three metrics which exhibited changes over field seasons).

Whilst all four female chimpanzees took longer to crack nuts in later years, the extent of change varied across individuals, from profound changes that are more likely to indicate senescence to changes that were more likely attributable to small-scale chance fluctuations across nuts (see *Figure 4*). Yo exhibited the largest increase in time between 1999 and 2016 (+28.4 s; difference between 1999 and 2016 = +104%; see Appendix 4 for an additional dataset for the duration of nut cracking behaviors performed by Yo in 2018, and corresponding analysis, which demonstrates that Yo took longer to crack oil-palm nuts at an even older age), and Velu exhibited the second largest increase in total time taken to crack oil-palm nuts (+26.5 s; +158%). Jire exhibited a more moderate increase in time taken to crack oil-palm nuts during the same timeframe (+9.7 s; +79%), followed by Fana, who exhibited the smallest change across all females (+5.7 s; +52%). For Fana, it is unclear whether this change represents chance fluctuations, given such small changes in nut cracking duration. Tua, the adult male, exhibited little evidence of a change in oil-palm nut cracking duration across field seasons, with his mean duration of oil-palm nut cracking decreasing by 0.67 s by 2008 (the latest year Tua was observed cracking oil-palm nuts).

Between 1999 and 2016, the median number of actions Yo used to crack and process each oil-palm nut increased (+14 actions per nut; +108% change between 1999 and 2016), as did the median

number of times Yo struck each nut with the hammer stone (+8 strikes per nut; +200% change between years). By analyzing the composition of Yo's action repertoire between 1999 and 2016 (*Appendix 2—table 10*), we determined that, in addition to more striking actions, Yo peeled shell away from kernels using her teeth more frequently in later years (peelteeth SHELL = +6.78% of the total action repertoire composition) and also consumed all kernel in a greater number of bites (eat KERNEL = +5.16% change in total action repertoire composition). Similarly, Velu exhibited an increase in the median number of actions used to crack and process each nut between 1999 and 2016, though this effect was milder (+7 actions; +70% change between 1999 and 2016). For Velu, the majority of these additional actions were strikes of the hammer stone (+4 strikes; +100%). For Fana and Jire, we detected a smaller change in the median number of actions used to crack open oil-palm nuts by 2016 (Fana = +2.5 actions; Jire = +3 actions); however, given that these changes are small, it is more difficult to rule out the possibility that they are due to chance fluctuations in samples between years. Additionally, for Fana and Jire, the median number of striking actions performed per nut in 1999 and 2016 was very similar (Jire: +2 strikes; Fana +1 strike). We found no evidence of Tua performing more or fewer actions when cracking oil-palm nuts (including hammer strikes) over field seasons.

## Discussion

We provide the first account of how a tool-use behavior performed by wild chimpanzees changed with progressively older age. Across a seventeen-year window, during which chimpanzees aged from 39-44 years to 56–61 years old, elderly chimpanzees began visiting experimental nut-cracking sites

**Table 2.** Summary of changes observed in each chimpanzee with progressive aging.

Summaries describe the differences between the first and last field season each individual was sampled (although models underlying each result used data from all field seasons). The term 'Possible Mild Increase/Decrease' is used to note where we identified a change for a particular metric, but this change was considerably smaller than for other individuals, and therefore could be due to chance. We address these instances on a case-by-case basis within the Results. Dashed lines (-) indicate where we found no strong evidence for behavioural change. Names of males are listed in all capitals; females' names are in capitals and lower case.

| Individual | Sex | Attendance | Engagement | Tool Selection Time | Oil-Palm Nut Cracking | | | Summary |
| | | | | | Total Time | # Actions | # Strikes | |
|---|---|---|---|---|---|---|---|---|
| Fana | F | Decrease | Decrease | Increase | Possible mild increase | - | - | Rate of attendance at the outdoor laboratory decreased, and Fana engaged with nuts and stones less often when present. Fana took longer to select tools. However, her general tool-using efficiency was mostly constant. |
| Jire | F | Decrease | Possible Mild Decrease | Increase | Mild increase | - | - | Rate of attendance at the outdoor laboratory decreased. Jire took longer to select stone tools. General tool-using efficiency was mostly constant, but Jire took slightly longer to crack oil-palm nuts in later years. |
| TUA | M | - | Possible Mild Increase | Decrease | - | - | - | Tua became slightly faster at selecting tools, and his tool-using efficiency was constant across years. |
| Velu | F | Decrease | Decrease (although sample size was small in 2011) | - | Increase | Increase | Increase | Rate of attendance at the outdoor laboratory decreased. Velu engaged with nuts and stones less often when present. Velu took longer to crack oil-palm nuts and used more striking actions. |
| Yo | F | Decrease | Possible Mild Increase | Increase | Increase | Increase | Increase | Rate of attendance at the outdoor laboratory decreased. When present, Yo spent a slightly greater proportion of time engaging with nuts and stones. Yo took much longer to select tools and demonstrated the largest reduction in tool-using efficiency. |

less frequently. This change in attendance rate was not observed for younger chimpanzees and was therefore confined to individuals who were approaching their maximum age in the wild. In addition, when present at nut cracking sites, two individuals (Fana and Velu) exhibited a lower engagement with nuts and stone tools in 2011 (the latest field season for this analysis) as compared with their behavior in earlier field seasons. With progressive aging, three individuals took longer to select stone tools (Yo, Fana, and Jire), several individuals took longer to crack open oil-palm nuts and consume all of the kernel, and two individuals (Yo and Velu) cracked and processed oil-palm nuts using a greater number of actions, including more frequent strikes of the hammer stone. Across metrics of engagement and efficiency, we detected interindividual differences in the extent to which nut cracking behaviors changed with progressive aging, including some individuals who exhibited little-to-no change in behavior across the time period sampled (see *Table 2*). Overall, our results provide initial evidence that elderly wild chimpanzees are subject to variable effects of aging on their habitual stone tool-use behaviors.

Whilst the outdoor laboratory locations are experimentally created nut-cracking sites, previous research on the performance of nut cracking across the home range at Bossou established that chimpanzees visit the outdoor laboratory at rates comparable to naturally occurring sites (*Almeida-Warren et al., 2022*), meaning that for most individuals at Bossou, the outdoor laboratory is a readily visited site for foraging. However, at progressively older ages, elderly chimpanzees began to visit the outdoor laboratory less frequently. This result was not found for younger chimpanzees between the ages of 8 and 30 years old, suggesting that changes in elderly chimpanzees' attendance rates were not an artefact of environmental changes over field seasons (which would likely affect both age cohorts similarly).

In addition, by 2011, two elderly chimpanzees (Velu and Fana) engaged with nuts and stone tools less frequently than they had done during visits to the outdoor laboratory throughout earlier years (although note for Velu, data for 2011 were collected from a single encounter). The possible causes for these changes in attendance rate – and engagement with nuts and stone tools when present at the outdoor laboratory – are manifold. Firstly, physiological changes with aging – such as in metabolism and nutrient requirements – could influence individuals' dietary preferences, reducing the appeal of nuts as an available food source. Secondly, physical senescence may make visiting nut cracking sites more challenging for elderly individuals, such as through reduced capacity for locomotion. Changes in locomotion have been documented in aging captive apes (*Neal Webb et al., 2019*). In the context of wild apes, reduced mobility may further restrict the available area that elderly chimpanzees can traverse during daily ranging, encouraging them to visit foraging patches that are more proximal to their immediate locations than the outdoor laboratory. Chimpanzees habitually plan their routes towards tool-use sites (*Almeida-Warren et al., 2022*); however, the extent to which their doing so changes during old age is unclear. Thirdly, it is possible that tool use may become more challenging due to changes in elderly chimpanzees' cognitive or physical condition (see latter sections of our discussion), which may reduce the appeal of visiting nut-cracking sites. Fourthly, changes in social association may have influenced the likelihood that elderly chimpanzees visited experimental nut-cracking sites. When experiencing increasingly old ages, many primates (*Neal Webb et al., 2019*; *Siracusa et al., 2022b*; *Campos et al., 2024*; *Almeling et al., 2016*; *Machanda and Rosati, 2020*) – and other mammals (*Siracusa et al., 2022a*; *Rudd et al., 2024*; *Albery et al., 2022*) – display higher social selectivity and spend a greater proportion of their time either in isolation or in smaller group associations. This social aging arguably may not be senescence per se, but can either be the result of senescent processes (perhaps even exacerbating senescence; *Siracusa et al., 2022a*), or may act to benefit individuals experiencing senescence, such as reducing the risk of injury from antagonistic interactions with social partners, or by shielding oneself from contagious disease (*Siracusa et al., 2022a*; *Siracusa et al., 2024*). In the context of our study, chimpanzees at Bossou typically arrive at the outdoor laboratory when travelling in groups, and therefore party members may be led to nut-cracking sites by associated conspecifics, rather than through independent choice. Previous research has demonstrated that whilst gregariousness appears to be constant across ages in the Bossou chimpanzees (*Schofield et al., 2023*), two old-aged individuals became notably peripheral in social networks constructed using data from the 2011 field season (*Schofield et al., 2019*) (Yo and Velu; the two individuals we identified as having the lowest attendance rates), indicating that they were more often seen arriving at the outdoor laboratory alone or travelling as a pair. In tandem with the profound reduction in population size at Bossou throughout our study period (*Matsuzawa et al., 2004*), changes in social

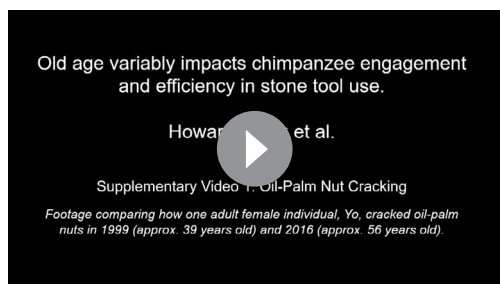

**Video 1.** Oil-palm nut cracking by Yo in 1999 and 2016. https://elifesciences.org/articles/105411/figures#video1

structure may have disproportionately affected elderly chimpanzees, leading to a lower likelihood of visiting the outdoor laboratory alongside – or to regroup with – other community members. Further research is required to link our observed changes in nut cracking engagement to their underlying causes.

Through the collection of longitudinal behavioral data, we were able to evaluate whether at older ages, chimpanzees perform less streamlined tool-use behaviors. At Bossou, chimpanzees exhibit interindividual variation in nut cracking efficiency; however, within-individual efficiency is stable over the majority of each chimpanzee's lifetime (*Berdugo et al., 2025*). To assess for aging effects, we therefore compared behavior in middle and later adulthood for each chimpanzee, permitting signals of aging to be identified using individually bespoke baselines of efficiency. Focal chimpanzees exhibited high efficiency in the first field season (1999), suggesting that we began sampling data before the onset of tool-use senescence (see *Biro et al., 2003* for data on striking efficiency of adults at Bossou, where all adults use approximately 2–4 strikes per oil-palm nut – our focal subset of adults fell within this range in 1999). Of the five individuals included in our study, three individuals took longer to select stone tools over progressing field seasons (Yo, Fana, and Jire), two females showed very small increases in the amount of time taken to crack and process nuts (Fana and Jire), whereas two other females exhibited comparatively dramatic increases in the time taken to crack and process each nut (Yo and Velu). Yo and Velu also exhibited more frequent striking using the hammer stone, and Yo performed additional actions to peel shell away from kernels with her mouth and consumed kernels with a greater number of bites (see *Video 1* for footage of Yo cracking oil-palm nuts in 1999 and 2016). Our results therefore suggest that there is measurable interindividual variation in the effects of extreme aging on the efficiency of both tool selection and use in wild chimpanzees.

Our findings mirror reports of interindividual variability in the senescence of cognitive and physiological systems in both captive and wild primates (*Tarou et al., 2002*; *Rothwell et al., 2021*; *Morbeck et al., 2002*; *Freire-Cobo et al., 2021*; *Hämäläinen et al., 2015*). Whilst it is outside the scope of our study to determine the precise cognitive, physical, and resultant motivational changes that led to reductions in engagement with tools, rates of tool selection, and the efficiency of tool use, our results can generate predictions about how changes in fine-scale processes may translate into the behavioral changes we have observed with progressive aging. For the chimpanzees that experienced reductions in nut cracking efficiency (as discussed above), these changes could be due to changes in physical strength and dexterity (*Lowenstine et al., 2016*; *Morbeck et al., 2002*), visual acuity (*Fujisawa et al., 2010*), dentition (where tooth decay may render peeling and chewing actions more difficult; *Lowenstine et al., 2016*; *Albrecht et al., 2024*), or possibly due to changes in cognition relevant to effective tool use (*Lacreuse et al., 2014*; *Lacreuse et al., 2020*), such as executive functioning or working memory. Changes in tool use behavior may also have emerged to actively compensate for the effects of senescence that are not in themselves specific to tool use; for example, tooth wear and subsequent periodontal disease are common in old-aged chimpanzees (*Lowenstine et al., 2016*; *Albrecht et al., 2024*). The additional strikes of the hammer stone performed by Yo and Velu to crack oil-palm nuts may have therefore been motivated by the desire to break the kernel into a greater number of smaller pieces, which were easier to peel and consume. This conclusion is somewhat supported by post-mortem data for Velu following her death in 2017, as she exhibited heavy wear patterns on both incisors and premolars and was missing several molars on the lower jaw (*Matsuzawa, 2018*). Similarly to tool-using efficiency, the duration of tool selection may have taken longer for several individuals due to difficulties identifying the properties of available stones, perhaps due to changes in perceptual systems (such as poorer vision; *Fujisawa et al., 2010*), or cognitive challenges in predicting the properties of objects. Alternatively, longer tool-selection times could have also been due to the need for chimpanzees to weigh up the benefits of specific tool properties (e.g. weight and size), relative to age-related changes in their physical strength and mobility. Bridging the gap between age-related

changes in tool-use behavior and changes in cognitive and physiological processes requires further study and would likely benefit from experimental approaches where possible.

Our study used longitudinal video data collected from a unique, decades-long systematic study of wild chimpanzee tool use, and this video archive likely represents the only currently available data source to study intraindividual variation in chimpanzee nut cracking across the span of decades (*Berdugo et al., 2025*). Moreover, given the exceptionally long lives of several chimpanzees at Bossou (*Emery Thompson et al., 2007*), Bossou represents a unique population whose individuals can be readily studied to examine the effects of senescence in the wild at the level of the individual. However, the use of existing video data and time-intensive methods for fine-scaled behavior coding means that our study faced a number of limitations.

Firstly, we were unable to analyze whether specific ecological variables – such as the availability of different food sources and demographic changes at Bossou – influence aging individuals' attendance at the outdoor laboratory, as well as at naturally occurring nut cracking sites (see discussion above). Nevertheless, by using the attendance of younger individuals as a control, we were able to identify that the reduction in attendance over field seasons was limited to older individuals, suggesting a specific change in behavior associated with chimpanzees experiencing progressively old age. Further research is needed to understand what factors lead to this reduced engagement with nut cracking at older ages; how aging individuals compensate for calorie loss from reduced nut cracking, and more generally, how aging influences chimpanzees' foraging behaviors across their entire home range.

Secondly, we were not able to eliminate all possible ecological explanations for age-related changes in tool-using efficiency (*Falótico et al., 2022*; *Proffitt et al., 2022*). Some ecological variables were controlled for as part of the experimental set up at the outdoor laboratory. For example, chimpanzees were always tested during similar months of the year in each field season, and were provided with oil-palm nuts at a suitable stage of maturity (and thus of similar hardness). Previous studies at Bossou have also revealed that chimpanzees are able to select nuts that are suitable for cracking, further suggesting that chimpanzees can account for differences in nut quality during tool use (*Sakura and Matsuzawa, 1991*; although this could also have been affected by aging itself, such as through reduced visual acuity impeding suitable nut selection; *Fujisawa et al., 2010*). Whilst it is possible that other ecological factors may influence our results, we believe that inter-seasonal ecological differences are unlikely to fully explain them. Firstly, if differences in chimpanzees' nut cracking efficiency between seasons were being driven by ecological variation, we predict that this inter-annual variation would be equally present across early and late field seasons. However, contrary to this prediction, nut-cracking efficiency was very similar across early field seasons (and across individuals during these years); changes in efficiency were only detected in later field seasons, when chimpanzees were sampled at older ages. Secondly, if inter-annual differences in ecology rendered nut cracking more difficult in later field seasons, we would expect these ecological changes to affect all individuals similarly. This was also not the case, as changes in behavior differed between individuals – some maintained similar levels of efficiency (such as Fana), whereas other individuals experienced profound reductions in efficiency (such as Velu and Yo). For Yo (the individual with the greatest reduction in tool-using efficiency), the observations that Yo took even longer to crack oil-palm nuts in 2018 (and that Yo also struggled to crack coula nuts in 2011, see Appendices 3 and 4 for further information, and *Video 2*) provide evidence for a directional, individual-specific reduction in nut-cracking efficiency across later years, rather than her efficiency being dictated by ecological variation. We could not provide nut-cracking efficiency data for a cohort of younger adults across field seasons, as there was only one young adult (male) individual at Bossou between 1999 and 2016. An additional younger adult cohort would have been a desirable control group for our study, to complement the long-term efficiency baselines we provide at the level of the individual. For future studies, we believe that the collection of a wider array of ecological data would be valuable for discriminating between explanations with

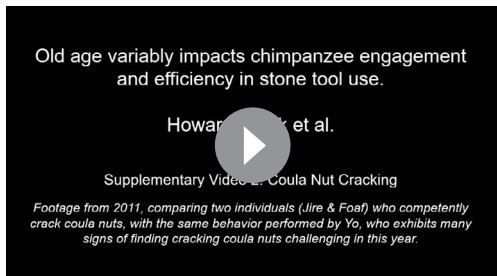

**Video 2.** Coula nut cracking by Jire, Foaf and Yo in 2011.
https://elifesciences.org/articles/105411/figures#video2

greater confidence and, where possible, future studies should aim to collect long-term data on the behavior of younger adults as additional control groups.

Thirdly, as data were collected from video footage, we could not examine how the dimensions of tools and raw material types influenced chimpanzees' tool-use behaviors over different encounters and across field seasons. Throughout long-term data collection at Bossou, the stone tools available at the outdoor laboratory have been kept as consistent as possible, and chimpanzees select tools with physical properties which likely enhance nut cracking efficiency (*Carvalho et al., 2008*; *Braun et al., 2025*). Indeed, changes in tool-using efficiency may be borne from increasing difficulty selecting appropriate tools with aging. Future studies should gather data on tool properties to support analyses of aging effects in tool use. These studies could be combined with data collection on even more finely grained behaviors of apes when using tools, such as the specific grips used to manipulate objects over time (*Neufuss et al., 2017*; *Malherbe et al., 2024*).

Fourthly, our study was limited to a small sample size of five aging chimpanzees. This is an inevitability of collecting longitudinal data on long-lived wild individuals who are approaching the ends of their lifespan. Further data is needed – including from other field sites – to establish whether the variation in aging effects we have found generalizes to other individuals and populations (and, where possible, other tool use behaviors).

Much like human technical skills, the tool-use behaviors of chimpanzees and other great apes are culturally variable behaviors that are acquired through a mixture of social learning and individual practice (*Matsuzawa et al., 2011*; *Inoue-Nakamura and Matsuzawa, 1997*; *Biro et al., 2003*; *Schuppli et al., 2016*). As previously mentioned, prior research on nut cracking behavior at Bossou has revealed that chimpanzees vary in the age of skill acquisition (*Matsuzawa et al., 2011*; *Inoue-Nakamura and Matsuzawa, 1997*), and the efficiency with which nut cracking is performed across adulthood varies between individuals, yet is stable within individuals over time (*Berdugo et al., 2025*). These differences in individual performance are, in part, likely due to differences in the environments of learners during skill acquisition (*Berdugo et al., 2024*). Our results expand on these findings to highlight that, whilst chimpanzees at Bossou perform nut cracking behaviors throughout their lives (suggesting that this is an important skill for this population), progressive aging and senescence may exaggerate interindividual differences in the performance of socially learnt skills, as seen in humans (*Lindenberger, 2014*; *Paccagnella, 2016*). Taken as a whole, our results provide further evidence that the cultural behaviors of humans and chimpanzees exhibit marked similarities that extend across their lifetimes, including, for some individuals, a period of technological senescence that can manifest across the oldest years of life. Further research should examine whether similar factors influence senescence of technical skills between humans and apes. This research should include studies that determine whether skill proficiency in early life and adulthood, and the frequency with which skills continue to be performed in later life, influence the rate and likelihood of senescence. Furthermore, debate continues as to whether human behaviors are particularly susceptible to the effects of aging due to the development of neurodegenerative diseases, such as Alzheimer's and Parkinson's disease (*Edler et al., 2021*; *Finch and Austad, 2015*; *Blesa et al., 2018*; *Emborg, 2017*). Current evidence suggests that the effects of these neurological changes in humans have far more profound impacts on behavior, despite primates sometimes exhibiting similar neuropathological signatures during aging (*Frye et al., 2022*; *Arnsten et al., 2019*; *Edler et al., 2017*; *Lemere et al., 2008*). Yet, it is still not clear whether this conclusion comes from a lack of behavioral data on aging in primates. We were not able to collect neurological samples from the chimpanzees in our study and cannot say whether the changes we observed could be associated with such processes (including for Yo, whose more profound senescence renders her the most likely to suffer from neurological decline). However, tool use could offer a suitable domain in which to examine the possible manifestation of such diseases (if present). Field sites should make specific efforts to collect long-term data on behavioral changes with aging – including for tool use – and where possible aim to sample neurological data following the natural deaths of wild, elderly individuals. These data will provide important insights into the underlying causes of human and chimpanzee skill disruption in the contexts of healthy and diseased aging.

In foraging contexts, multi-object tool-use behaviors are often used to extract energy-rich resources that would not be otherwise accessible (*King, 1986*). Given the sample size of our study and limitations in the wider ecological data available, it was not possible to identify the influence of tool-use senescence on individual survivability through reduced energy intake. Answering this question would

require data on a greater number of tool-use behaviors, spanning across the home range. However, the two females in our study who exhibited limited changes in tool-using efficiency (Jire and Fana) had the longest lifespan after the end of our study, whereas Velu and Yo (who experienced greater difficulty cracking nuts in later years) died comparatively shortly after our sample period. While this sample is too small to make strong claims, this pattern could indicate that the senescence of tool-use behaviors, and the energy-rich foods to which they permit access, can contribute to reduced surviv-ability in the wild. Alternatively, changes in tool-use behaviors could offer a litmus for identifying individuals experiencing profound generalized senescence in the wild, where changes in many phys-iological and behavioral processes subsequently impact an individual's survivability. By extension, outside of tool use, our data on individual survivability – alongside previous studies of chimpanzees at Bossou – invites speculation surrounding whether the social relationships of specific individuals may reduce or accentuate the rate and intensity of senescence with progressive aging. In addition to living for a longer period after the final field season we sampled, Fana and Jire enjoyed more central social positions than other elderly females at Bossou (*Schofield et al., 2019*) and had a greater number of living relatives during the latter field seasons (during which Jire had two mature offspring, and Fana had two mature offspring and two young grandchildren). Given that only eight individuals remained at Bossou by 2016, these familial relationships constituted a significant proportion of the overall community. Further research will provide insight into if and how lifetime sociality and familial success influence the onset and rate of senescence in wild chimpanzees, for example through socially mediated resource acquisition.

In sum, we demonstrate that the tool-use behaviors of wild chimpanzees can change in old age, and that these changes vary between individuals. Whilst our study is observational, we argue that processes of senescence offer a promising explanation for such changes, particularly in regard to changes in individual tool-using efficiency. Our findings reflect the conclusions of previous studies investigating the cognitive and physiological senescence of primates living in captivity and generate a myriad of further questions about the possible underlying mechanisms of tool-use senescence, as well as more general behavioral aging in wild apes. As questions surrounding aging benefit from longi-tudinal data – which for many primates require collection across decades – the possibility that such questions will be answered is becoming increasingly more precarious in an age where primates are at a heightened risk of extinction (*Estrada et al., 2017*). The conservation of wild primate populations is paramount for our understanding of the dynamics and evolution of primate aging processes, which may reveal insights into the forces influencing behavioral stability and senescence across the human lifetime.

## Methods

### Study site and video archive

Established in 1976, Bossou (07° 390' N; 008° 300' W) is a site of long-term research focused on a small population of wild West-African chimpanzees (*Matsuzawa et al., 2011*). Chimpanzees at Bossou possess an extensive repertoire of tool-use behaviors (*Humle, 2011*), including nut cracking (*Carvalho et al., 2008*; *Matsuzawa, 1994*; *Biro et al., 2003*; *Biro et al., 2006*). For many years, the size of the Bossou chimpanzee community remained relatively stable at around 20 individuals; however, following a flu-like epidemic in 2002 that killed 5 individuals (*Matsuzawa et al., 2004*), the population began an irrecoverable decline. At the time of writing (2025), the Bossou community consists of just three individuals.

Nut cracking by Bossou chimpanzees has been studied for several decades through the use of an 'outdoor laboratory' (*Matsuzawa, 1994*; established in 1988; see *Figure 1*). Open during the dry season of each year (approximately November-February), the outdoor laboratory is a maintained natural clearing in the forest (approximately 7x20 m) where nuts and stones are provided for chimpan-zees. These typically consist of oil-palm nuts, *Elaeis guineensis,* which is the only species that occurs naturally at Bossou and thus the only species of nut habitually cracked by these chimpanzees. Oil-palm nuts were sourced from local communities surrounding Bossou, and nuts provided to chimpanzees were generally of suitable maturity for cracking and consumption. In some years, coula nuts, *Coula edulis,* and in 1 year panda nuts, *Panda oleos*, were also provided. Neither coula nor panda nuts occur naturally at Bossou, but both are cracked by chimpanzees at nearby sites (*Biro et al., 2003*). In

addition to nuts, chimpanzees are also provided with over 50 stones organized in a square matrix in the center of the clearing at the start of each session (see *Figure 1d*), as well as a raceme of edible oil-palm fruits (see *Figure 1c*). The outdoor laboratory also features a water point (a hollowed section of a tree trunk; see *Figure 1b*) from which chimpanzees may extract water for drinking using leaf tools. Chimpanzees visit the outdoor laboratory spontaneously as part of their daily ranging behaviors, and with comparable frequency to other naturally occurring nut cracking sites within the home range at Bossou (*Almeida-Warren et al., 2022*). Upon arrival at the outdoor laboratory, chimpanzees' behaviors are recorded by observers using two or three video cameras located behind a screen of vegetation.

Video data collected at the outdoor laboratory has recently been collated into a longitudinal archive spanning from 1990 to the present day (*Berdugo et al., 2025*; *Schofield et al., 2019*; *Bain et al., 2021*). Given the long-term use of the outdoor laboratory to study nut cracking, all chimpanzees were well habituated to the experimental set up by 1999 (the first field season from which data was sampled for this study). The Bossou archive also contains data on the composition of all recorded encounters with groups of chimpanzees at the outdoor laboratory across this timeframe and can be combined with recorded family lineage data (including birth data for individuals born after the establishment of long-term research at Bossou). The Bossou archive therefore presents a novel opportunity for longitudinal analysis of whether and how tool-use behaviors change across the later periods of chimpanzee lifespans.

We collected and analyzed data on the behaviors of chimpanzees at the outdoor laboratory at Bossou, spanning a seventeen-year period (1999–2016). This 17-year period contains the majority of the available video data collected at Bossou. To balance fine-grained behavior coding (which is highly time intensive) with the need to sample data over an extended time period to capture possible effects of long-term senescence, we aimed to sample field seasons within this time window at 4-year intervals; however, in some years, data was not collected at Bossou (such as during years where disease outbreaks were affecting local communities). In such instances, we sampled the closest field season which likely contained sufficient data for our analyses. Thus, during this period, we sampled data from five field seasons (1999, 2004, 2008, 2011, and 2016).

Over the years, the outdoor laboratory has operated at two locations in the forest. Between 1999 and 2008, data was only collected at one specific location across the years. However, in 2011, a second clearing was made available to chimpanzees, with an identical experimental set up. In 2011, data was collected in both locations simultaneously. By 2016, all data collection had transitioned to the newer, second location. For the majority of our analyses, we therefore used data from the first location between the years of 1999–2011 and the second location in 2016. For only one analysis (attendance rate, see below), we used data from both the first and second outdoor laboratory location in 2011. All behavior coding in our study was performed using the open-access software BORIS (*Friard and Gamba, 2016*), with videos viewed on a 59 cm monitor.

## Attendance

We recorded the total number of times chimpanzees were encountered at the outdoor laboratory in each field season. An encounter was defined as any instance where one or a group of chimpanzees visited the outdoor laboratory and lasted until all chimpanzees left. If a chimpanzee left the outdoor laboratory and returned before all other individuals had left, it was classed as part of the same encounter. We collected data on the five chimpanzees experiencing increasingly old age, as well as for an additional cohort of all adult and subadult individuals between the ages of 8 and 30 years old. This second cohort of individuals allowed us to evaluate whether changes in attendance rate over progressive field seasons may be due to confounding factors other than senescence, such as changes in environmental food availability or population structure, or changes in the number of experimental nut cracking sites. For our younger cohort of chimpanzees, we used a minimum age of 8 years old, as chimpanzees at Bossou acquire the skill of nut cracking by 7 years of age. For this analysis, we also excluded two adult females (Pama and Nina) who never learnt to successfully crack nuts using stone tools. To evaluate whether progressive aging reduced attendance rates, we constructed a Poisson GLMM with field season as a continuous fixed effect (scaled about the mean), and cohort (above or below 30 years) as a categorical variable. Rates of attendance were estimated using encounters (a count variable) and the inclusion of an offset term in our model (the number of days of the field

season). Importantly, we included an interaction term in our model to see whether the relationship between aging across field seasons and attendance rate was more severe for our focal elderly individuals compared with the younger cohort. Individual ID was included as a random intercept to account for repeated measures of the same individual across field seasons.

## Behaviors at the outdoor laboratory

We estimated whether, over successive field seasons, elderly individuals began spending a greater or lesser proportion of their time engaging in nut cracking when present at the outdoor laboratory, as compared with other commonly performed behaviors. For each elderly individual, we sampled the first 10 encounters of each field season in which they were present (totaling a maximum of 50 encounters per individual across all field seasons). For each encounter, we timed how long individuals engaged in one of four mutually exclusive categories of behavior:

1. **Engaging with nuts and stone tools:** the total amount of time individuals spent interacting with nuts, nut fragments (including shells and kernels), and stones. Coding began when an individual first touched a nut, nut fragment, or stone.
2. **Drinking water:** the total amount of time individuals spent producing leaf tools and drinking from the water point. Coding began when an individual stripped leaves from branches to begin manufacturing a leaf tool to aid in drinking water, or, if reusing a tool that had previously been made and discarded, when they collected the tool and began to approach the water point to drink.
3. **Eating oil-palm fruits:** the total amount of time individuals spent consuming oil-palm fruits from the raceme. Coding began whenever an individual started picking oil-palm fruits from the raceme, or collecting and consuming oil-palm fruits off of the ground.
4. **Other:** any other behaviors (e.g. grooming, playing, resting).

For behaviors 1–3, coding ceased when individuals stopped interacting with all relevant objects of one behavior (e.g. leaf tools, stones, fruits) and began engaging in another behavior (either another behavior from 1 to 3, or 'Other' behaviors such as grooming). If an individual dropped all relevant items and was idle for at least 1 min, they were considered to be resting, and behaviors were marked as 'Other'. For field seasons in which individuals did not visit the outdoor laboratory on at least 10 occasions, all available encounters were used. We also recorded the total amount of time each focal individual spent at the outdoor laboratory during each encounter. We calculated the proportion of time individuals spent engaging in each type of behavior across all encounters of each field season (thus, controlling for differences in the total time individuals were observed between years). For this analysis, we omitted any videos collected at the second outdoor laboratory site (including some videos from 2011 and all videos from 2016). We chose to do so as the two sites vary in how accessible they are (with the original site being situated at the top of a steep hill, but the second requiring no climb). This difference in terrain may influence the behaviors of the chimpanzees once arriving at the outdoor laboratory, for example, more time resting following climbing.

## Stone-tool selection

We measured the time taken for the old-aged chimpanzees to select stone tools at the central matrix during each of the sampled field seasons. We sampled the first 10 encounters with each individual for each field season. All tool-selection events in these encounters were sampled. We began timing when chimpanzees approached the matrix, and we considered their gaze to be fixed on at least one stone tool. We stopped timing when chimpanzees turned away from the matrix holding all acquired stone tools. We recorded the number of stone tools taken by the focal chimpanzee during each instance of stone-tool selection (a categorical score of 'one' or 'two' stones). Additionally, we recorded the number of stone tools that had been removed from the stone-tool matrix prior to each stone selection event (including those removed by other individuals, as well as the focal chimpanzee during previous instances of stone-tool selection). We could therefore incorporate into our analysis a metric of how many options were available to chimpanzees during each stone-tool selection event (where higher numbers of previously selected tools indicated a smaller pool of possible choices). We did not sample instances where chimpanzees selected and used tools which were not positioned at the central matrix (i.e. when using tools which had already been transported away from the matrix by other chimpanzees). When modeling tool-selection time over field seasons, both the number of stones

that a chimpanzee selected from the central matrix (one or two) and the number of stones previously removed from the matrix were included in our model as fixed effects, to control for additional variation in selection time introduced by these two factors (see Statistical analyses below for more details on GLMM model structures).

## Coding discrete actions

When performing complex technical skills—including tool-use behaviors— experienced apes organize individual actions into more streamlined goal-directed sequences (*Byrne et al., 2013*; *Howard-Spink et al., 2024*; *Inoue-Nakamura and Matsuzawa, 1997*; *Boesch et al., 2020*), and begin to successfully perform behaviors more quickly (*Howard-Spink et al., 2024*; *Boesch and Boesch, 1983*; *Biro et al., 2003*; *Neufuss et al., 2017*); however, whether experiencing progressively old age influences this efficiency has not yet been studied.

To evaluate the efficiency with which focal chimpanzees engaged in nut cracking, we coded the individual, fine-grained actions used by focal chimpanzees when cracking nuts at the outdoor laboratories. We sampled videos chronologically from each sampled field season. Fine-grained action coding was conducted for each individual in each field season until at least 1000 actions had been collected, and included sequences of actions that described the cracking of at least 20 nuts in their entirety. To code sequences of actions, we used an ethogram of 34 manipulations (e.g. 'Grasp', 'Drop', 'Strike') and 6 objects (see *Appendix 1—table 1* for full ethogram; all discrete-action coding was performed by one individual: EHS; our ethogram was tested to ensure that it can be applied objectively as part of a previous research project; *Howard-Spink et al., 2024*). The six possible objects included: Hammer, Anvil, Nut, Kernel (the edible interior of a nut), Shell (the inedible exterior of a nut), and Bare Hand (for use when individuals performed an action in the absence of an object, e.g., striking the anvil with bare hand). Stone tools were categorized as 'Hammer' or 'Anvil' based on how they were used by the individual. If an individual swapped over their hammer and anvil stones, the stones were reassigned accordingly. Thus, stone tools were coded according to their function at any specific point in a sequence. We considered an individual action to be the combination of both the manipulation performed by the chimpanzee and the object being manipulated (e.g. 'Grasp Hammer'=one action).

Coding of action sequences started when an individual began interacting with a nut, nut fragment, or stone, and ceased when an individual dropped all relevant objects and either began engaging in an alternative behavior (e.g. grooming, feeding on oil-palm fruits, etc.) or rested for longer than one minute. Actions were coded as discrete events during behavioral observation, and the time of each action was automatically recorded in BORIS.

The total corpus of actions was parsed into sequences directed at the cracking of individual nuts and the consumption of all associated kernels. Only sequences which described the entire processing of a nut, from acquisition of the nut to complete consumption of the kernel, were included in our analysis. A number of metrics of efficiency were extracted from action sequences directed at individual nuts, which emphasize the speed with which kernels can be acquired and consumed, and the extent to which the actions used to do so reflect a streamlined behavioral sequence:

1. **Total time to crack nuts and consume all kernel.** The total time between the start of the first action and the end of the last action performed on a specific nut, including all actions involving the consumption of kernel and disposal of shell fragments.
2. **Number of actions used to process nuts and consume all kernel.** The total number of actions performed during the cracking of a specific nut, including consumption of kernel and disposal of shell fragments.
3. **Number of unique types of action used to process nuts and consume all kernel.** The number of different types of actions performed by an individual when cracking a specific nut, including all kernel consumption and disposal of shell fragments. This metric is different from [2] as it describes the variety of actions employed, rather than the number; for example, for the action sequence *A,A,B,C,B,C*, where letters denote hypothetical actions, the number of actions is six, and the number of unique action types is three {*A, B, C*}.
4. **Number of times the nut was (re)placed on the anvil.** The number of times the nut was placed on the anvil during initial placement, and also all replacements when the nut rolled off the anvil. An understanding of how to orient the anvil and where to place the nut can reduce the likelihood of needing to perform nut replacements. Therefore, fewer nut placements reflect a greater efficiency in nut cracking.

5. **Number of strikes of the hammer stone per nut.** The number of individual strikes of the hammer stone performed on each nut that a chimpanzee cracked. This is a common metric for efficiency in chimpanzee nut cracking (*Biro et al., 2003*; *Boesch et al., 2019*), where fewer strikes indicate higher efficiency.

6. **Number of tool reorientations per nut.** The total number of horizontal rotations ('reorient') and vertical rotations ('flip') of hammer and anvil stones. Chimpanzees occasionally reorient tools during tool use, with these adjustments likely reflecting attempts to achieve greater efficiency. Once a suitable configuration of tool objects is acquired, such reorientations are usually less common.

7. **Tool changes per nut.** The number of times an individual swapped over their hammer and anvil stones, swapped out at least one stone tool for a new stone, or moved across the outdoor laboratory to use a hammer-anvil set previously abandoned by another user.

## Statistical analyses

Statistical analyses were performed in R (version 4.3.3 Angel Food Cake). For attendance rates, tool-selection times, and the metrics of efficiency cracking individual nuts, we used linear and generalized linear mixed-effect models to assess for the effects of age (using the lme4 package; *Bates et al., 2015*). Linear mixed-effect models were used in any instance where time was a response variable (time was logged to confer normality); whereas, if a response variable was measured in counts (e.g. number of actions used to crack open a nut; number of encounters over a field season, etc.), we employed generalized linear mixed-effect models using a Poisson distribution. In both instances, the year of the field season (scaled about the mean) was used as a fixed effect, as a proxy for age.

Our model for attendance data is described above in **Attendance** (we describe this model separately as we modeled the effect of aging in two separate groups – young and old – rather than for specific individuals). For all other metrics, we modeled the effect of aging on focal, elderly chimpanzees by using random intercepts and slopes to evaluate the effect of the increasing year of each field season. Our decision to include a random slope is supported by previous research on captive and wild primates, which suggests that the effects of senescence are highly variable across individuals for a number of physiological and cognitive processes (*Tarou et al., 2002*; *Rothwell et al., 2021*; *Morbeck et al., 2002*; *Freire-Cobo et al., 2021*; *Hämäläinen et al., 2015*). Across metrics of engagement and efficiency, we compared random-slope models with fixed slope models – which assume that the effect of increasing field season will be the same for all individuals – as well as with null models which fit an intercept across all field seasons (and therefore assumed no effect of aging). This allowed us to evaluate whether the effect of increasing year of each field season influenced chimpanzee behavior, and also whether this effect was variable across individuals. All model comparisons were performed using AIC, where lower scores indicate better fit relative to model complexity. Where necessary, we also included the specific encounter as an additional random intercept (fixed slope) to avoid pseudoreplication from multiple tool-selection events within the same encounter.

When analyzing metrics of nut cracking efficiency (for which we had a total of seven metrics, see above), we restricted the use of models to describe changes in metrics based on whether, for at least one individual, the median value for a particular field season fell outside the interquartile range of any previous field season. If a model identified that an efficiency metric was changing across field seasons, we used model coefficients to gauge the direction of change for each individual. To estimate the magnitude of change across field seasons, we compared data collected for each individual in the earliest and latest year that they were sampled. We calculated % change as the change in a metric between the last and first field season, divided by the value for the first field season (to scale the change), multiplied by 100 (see associated code).

## Acknowledgements

We thank Daniel Schofield for their help in curating the Bossou video archive, as well as Ignacio Juarez Martinez and Alex Mielke for their statistical advice. For both assistance and permission to conduct research at Bossou, we thank the Direction Générale de la Recherche Scientifique et de l'Innovation Technologique (DGERSIT) and the Institut de Recherche Environnementale de Bossou (IREB), République de Guinée. For their invaluable help in the field, we would also like to thank the research assistants Boniface Zogbila, Gouanou Zogbila, Tino Zogbila, Henry Didier Camara, Marcel Doré, Jules

Doré, Pascal Goumy, Gouanou Goumy, Paquile Cherif, Cé Vincent Mamy, Dagouka Samy as well as all other guides. Additionally, we thank all KUPRI-International researchers (especially Misato Hayashi, Claudia Sousa, Kimberly Hockings, Katelijne Koops, Tatiana Humle, Gaku Ohashi, Hiroyuki Takemoto, Gen Yamakoshi, Yukimaru Sugiyama, and Jérémy Koman), and IREB researchers (especially Mamadou Diakité, Makan Kourouma, and Aly-Gaspard Soumah) who contributed to the wider program of data collection at Bossou during the period of our study.

## Additional information

### Funding

| Funder | Grant reference number | Author |
| --- | --- | --- |
| Natural Environment Research Council | NE/L002612/1 | Elliot Howard-Spink |
| Swiss National Science Foundation | PCEFP1_186832 | Thibaud Gruber |
| Templeton World Charity Foundation | 10.54224/20647 | Dora Biro |
| Ministry of Education, Culture, Sports, Science and Technology, Japan | #12002009 | Tetsuro Matsuzawa |
| Ministry of Education, Culture, Sports, Science and Technology, Japan | #16002001 | Tetsuro Matsuzawa |
| Ministry of Education, Culture, Sports, Science and Technology, Japan | #20002001 | Tetsuro Matsuzawa |
| Ministry of Education, Culture, Sports, Science and Technology, Japan | #24000001 | Tetsuro Matsuzawa |
| Ministry of Education, Culture, Sports, Science and Technology, Japan | #16H06283 | Tetsuro Matsuzawa |
| Japan Society for the Promotion of Science | Core-to-core CCSN | Tetsuro Matsuzawa |
| Japan Society for the Promotion of Science | U04-PWS | Tetsuro Matsuzawa |
| Fundação pela Ciência e Tecnologia (with support from Programa Operacional Capital Humano and the European Union) | SFRH/BD/115085/2016 | Katarina Almeida-Warren |
| The Boise Trust Fund, University of Oxford | | Katarina Almeida-Warren |
| The National Geographic Society | EC-399R-18 | Katarina Almeida-Warren |
| The Leverhulme Trust | ECF-2022-322 | Katarina Almeida-Warren |
| The Daiwa Foundation | | Catherine Hobaiter |
| European Union's 8th Framework Programme, Horizon 2020 | 802719 | Catherine Hobaiter |
| Ministry of Education, Culture, Sports, Science and Technology, Japan | MEXT | Tetsuro Matsuzawa |

| Funder | Grant reference number | Author |
|---|---|---|

The funders had no role in study design, data collection and interpretation, or the decision to submit the work for publication.

## Author contributions

Elliot Howard-Spink, Conceptualization, Data curation, Formal analysis, Funding acquisition, Validation, Investigation, Visualization, Methodology, Writing - original draft, Writing – review and editing; Tetsuro Matsuzawa, Resources, Data curation, Funding acquisition, Investigation, Methodology, Project administration, Writing – review and editing; Susana Carvalho, Catherine Hobaiter, Katarina Almeida-Warren, Data curation, Investigation, Writing – review and editing; Thibaud Gruber, Conceptualization, Supervision, Methodology, Writing – review and editing, Cosenior Author; Dora Biro, Conceptualization, Data curation, Supervision, Investigation, Methodology, Writing – review and editing, Cosenior Author

## Author ORCIDs

Elliot Howard-Spink ⓘ https://orcid.org/0000-0003-3961-3982
Tetsuro Matsuzawa ⓘ http://orcid.org/0000-0002-8147-2725
Susana Carvalho ⓘ http://orcid.org/0000-0003-4542-3720
Catherine Hobaiter ⓘ http://orcid.org/0000-0002-3893-0524
Katarina Almeida-Warren ⓘ https://orcid.org/0000-0002-7634-9466
Thibaud Gruber ⓘ http://orcid.org/0000-0002-6766-3947
Dora Biro ⓘ http://orcid.org/0000-0002-3408-6274

## Ethics

Data collection at Bossou has been conducted by different researchers over several decades. Research authorization and ethical approval were given to T.M. (who oversaw data collection during the study period covered by this project) through a memorandum of understanding between Kyoto University, and the Guinean authorities (Direction Générale de la Recherche Scientifique et de l'Innovation Technologique, DGERSIT, and the Institut de Recherche Environnementale de Bossou, IREB), to which all researchers conformed. The present study was entirely conducted using existing video data from Bossou (the Bossou Archive), and therefore did not involve collection of additional data from wild chimpanzees.

Reviewer #1 (Public review): https://doi.org/10.7554/eLife.105411.3.sa1
Reviewer #2 (Public review): https://doi.org/10.7554/eLife.105411.3.sa2
Author response https://doi.org/10.7554/eLife.105411.3.sa3

---

# Additional files

## Supplementary files
MDAR checklist

## Data availability
All data and code can be accessed through the open science framework: https://osf.io/uf9za/.

The following dataset was generated:

| Author(s) | Year | Dataset title | Dataset URL | Database and Identifier |
|---|---|---|---|---|
| Howard-Spink E | 2025 | Data: Old age variably impacts chimpanzee engagement and efficiency in stone tool use | https://osf.io/uf9za/ | Open Science Framework, uf9za |

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

# Appendix 1

## Ethogram for coding fine-scale object manipulations

**Appendix 1—table 1.** Ethogram of codable manipulations for observations of nut-cracking behaviors*.

Manipulations in bold are coded alongside a corresponding object. There are 6 possible corresponding objects: 1. Nut, 2. Hammer, 3. Anvil, 4. Kernel, 5. Shell, 6. Bare Hand. Codes in italics are used to denote the start and end of observable sequences. Coding commenced when individuals began interacting with stones, nuts, or nut fragments. Coding ceased when individuals began engaging in a new behavior, e.g. play, grooming, eating oil-palm fruits. On the occasions where an individual moved out of clear sight of the video recording, or behavior became obscured by an individual's body position, sequences were terminated with a 'Not Visible' codon and marked as incomplete. This ethogram has been used in previous studies to collect data from the Bossou archive, see *Howard-Spink et al., 2024*.

| Action | Description |
| --- | --- |
| Bite | Place object in mouth and bite with teeth. Differs from 'Eat' as there is no consumption. Differs from 'Store' as the object moves in and out of the mouth while the chimp is stationary. Differs from 'Peel Teeth' as bite applies general force, whereas peel with mouth is to remove shell fragments from kernel dexterously. Differs from 'Kiss' as the object enters inside the mouth. |
| Brush | Brush objects off the anvil. |
| Drop | Place object(s) on ground. |
| Eat | Consume object. Differs from bite and store as it requires successful ingestion. |
| Flip | Flip object over. |
| Grasp | Grasp an object and move off the ground. |
| Kick | Apply rapid, hard force on the object with the foot, so that the object is displaced. |
| Kiss | Place object to lips or nose, but not inside of the mouth. |
| Pass | Pass object between hands. |
| Peel Hand | Peel shell off of kernel with hands. |
| Peel Teeth | Peel shell off of kernel with teeth. |
| Place | Place an object on an anvil. |
| Provide | Directly hand an object to another individual. |
| Pull Foot | Pull an object across the ground with foot. |
| Pull Hand | Pull an object across the ground with hand. |
| Push Foot | Push an object across the ground with foot. |
| Push Hand | Push an object across the ground with hand. |
| Rake Foot | Pull many objects towards oneself using foot/leg. |
| Rake Hand | Pull many objects towards oneself using hand/arm. |
| Reorient | Rotate object horizontally. |
| Roll Foot | Roll object along the floor with foot. |
| Roll Hand | Roll object along the floor with hand. |
| Spit | Let object fall from mouth or lips. |
| Stomp | Whilst standing, apply strong force to object with foot. |

*Appendix 1—table 1 Continued on next page*

*Appendix 1—table 1 Continued*

| Action | Description |
| --- | --- |
| Store | Place object(s) in mouth for transportation. Separated from bite, eat, peel teeth as it is followed by transportation across the outdoor laboratory, before then being removed from the mouth intact. |
| Strike One Hand | Strike with one hand, (and associated object). |
| Strike Two Hand | Strike with two hands, (and associated object). |
| Support Foot | Support object with foot. |
| Support Hand | Support object with hand. |
| Take | Receive an object directly from another individual. |
| Throw | Throw object away from self, horizontally or vertically. |
| Touch Foot | Touch or grasp object without moving it using foot. |
| Touch Hand | Touch or grasp object without moving it using hand. |
| Relocate | Stand up and move to a new area; is immediately followed by another coding action. |
| *Start* | Start of a sequence. |
| *End Bout* | End of a sequence. |
| *Not Visible* | Individual moves out of view, and coding cannot continue. Differs from 'End Bout', as there is no evidence the individual has stopped engaging with stones, nuts or nut fragments. |

## Appendix 2

## Additional summary data and outputs of statistical models

**Appendix 2—table 1.** Model output for attendance rates over each year.

**glmer(Encounters ~Year-Scaled*Age_Cohort + (1| ID)+offset(log(Field_Season_Duration)), family = Poisson)**

| Random Effect | Variance | SD | | |
|---|---|---|---|---|
| ID (Intercept) | 0.055 | 0.235 | | |
| Fixed Effect | Estimate | SE | z value | p |
| (Intercept) | –0.570 | 0.107 | –5.35 | <0.001 |
| Year-Scaled | –0.123 | 0.078 | –1.57 | 0.116 |
| Old_Cohort | –0.198 | 0.141 | –1.40 | 0.160 |
| Year-Scaled:Old_Cohort | –0.220 | 0.096 | –2.29 | 0.022 |

**Appendix 2—table 2.** The number of stone-tool selection events sampled for each individual in each field season.

| Individual | Year | | | | | Total |
| | 1999 | 2004 | 2008 | 2011 | 2016 | |
|---|---|---|---|---|---|---|
| Fana | 2 | 2 | 8 | 0 | 4 | 16 |
| Jire | 4 | 4 | 5 | 3 | 3 | 19 |
| TUA | 6 | 9 | 7 | 3 | 0 | 25 |
| Velu | 3 | 5 | 10 | 0 | 3 | 21 |
| Yo | 4 | 5 | 8 | 9 | 1 | 27 |

**Appendix 2—table 3.** Models of stone-tool selection duration across years, with AIC.

AIC values reported for models fitted by Maximum Likelihood. PAB is the encounter in which stone tool selection events occurred.

| Model | Formula | df | AIC |
|---|---|---|---|
| REML = F | | | |
| Individual Random Slope and Intercept for Year-Scaled. | Log(Time)~Year-Scaled+Stones Selected + Stones Previously Taken + (1+Year-Scaled||ID) + (1|PAB) | 8 | 231 |
| Dropped Year-Scaled | Log(Time)~Stones Selected + Stones Previously Taken + (1|ID) + (1|PAB) | 6 | 234 |
| Null for All Fixed Effects | Log(Time)~1 + (1|ID)+ (1|PAB) | 4 | 234 |
| REML = T | | | |
| Random Slope & Intercept | Log(Time)~Year-Scaled+Stones Selected + Stones Previously Taken + (1+Year-Scaled||ID) + (1|PAB) | 8 | 244 |
| Fixed Slope & Random Slope | Log(Time)~Year-Scaled+Stones Selected + Stones Previously Taken + (1|ID) + (1|PAB) | 7 | 246 |

**Appendix 2—table 4.** Summary output for mixed-effect model for tool selection duration, with individual as a random slope, and encounter and individual as random intercepts.

108 observations, 49 encounters, 5 individuals.

**Log(Time)~Year-Scaled+Stones Selected + Stones Previously Taken + (1+Year-Scaled||ID) + (1|PAB), REML = F**

| Random Effect | Variance | SD |
|---|---|---|
| Encounter | 0.02 | 0.15 |

*Appendix 2—table 4 Continued on next page*

*Appendix 2—table 4 Continued*

**Log(Time)~Year-Scaled+Stones Selected + Stones Previously Taken + (1+Year-Scaled||ID) + (1|PAB), REML = F**

| ID (Intercept) | 0.13 | 0.35 | |
|---|---|---|---|
| ID (Slope – Year-Scaled) | 0.05 | 0.22 | |
| (Residual) | 0.35 | 0.59 | |
| Fixed Effect | Estimate | SE | t value |
| (Intercept) | 1.86 | 0.22 | 8.43 |
| Year-Scaled | 0.14 | 0.12 | 1.18 |
| Two Stones Selected | 0.31 | 0.15 | 2.14 |
| Stones Previously Taken | –0.01 | 0.02 | –0.65 |

**Appendix 2—table 5.** Individual coefficients for the mixed-effect model for stone-tool selection duration, with individual as a random slope, and encounter and individual as random intercepts.

| Individual | Intercept | Scaled Year (Slope) |
|---|---|---|
| Fana | 1.93 | 0.27 |
| Jire | 1.80 | 0.17 |
| TUA | 1.39 | –0.17 |
| Velu | 1.76 | 0.07 |
| Yo | 2.41 | 0.35 |

**Appendix 2—table 6.** Summary output for the random slope mixed-effect model for the total time taken to crack and process oil-palm nuts over sampled field seasons.
1538 observations, 20 encounters, 5 individuals. Values rounded to 3.d.p.

**lmer(Log(Time)~Year-Scaled + (Year-Scaled | ID) + (1| Encounter), REML = F)**

| Random Effect | Variance | SD | |
|---|---|---|---|
| Encounter | 0.029 | 0.169 | |
| ID (Intercept) | 0.089 | 0.298 | |
| ID (Slope – Year-Scaled) | 0.002 | 0.039 | |
| (Residual) | 0.418 | 0.647 | |
| Fixed Effect | Estimate | SE | t value |
| (Intercept) | 2.657 | 0.142 | 18.767 |
| Year-Scaled | 0.077 | 0.030 | 2.605 |

**Appendix 2—table 7.** Summary output for the random slope mixed-effect model for the number of discrete actions used to crack and process oil-palm nuts over sampled field seasons.
1538 observations, 20 encounters, 5 individuals. Values rounded to 3.d.p.

**glmer(Action Events ~Scaled Year + (Year-Scaled |ID) + (1|Encounter), family = poisson)**

| Random Effect | Variance | SD | |
|---|---|---|---|
| Encounter | 0.084 | 0.289 | |
| ID (Intercept) | 0.029 | 0.170 | |
| ID (Slope – Year-Scaled) | 0.003 | 0.059 | |
| Fixed Effect | Estimate | SE | z value |
| (Intercept) | 2.563 | 0.102 | 25.241 |
| Year-Scaled | 0.022 | 0.029 | 0.769 |

**Appendix 2—table 8.** Summary output for the random slope mixed-effect model for the number of strikes of the hammer stone used to crack and process oil-palm nuts over sampled field seasons.

1538 observations, 20 encounters, 5 individuals. Values rounded to 3.d.p.

**glmer(Strikes ~Year-Scaled + (Year-Scaled |ID) + (1|Encounter), family = poisson)**

| Random Effect | Variance | SD | |
|---|---|---|---|
| Encounter | 0.157 | 0.397 | |
| ID (Intercept) | 0.042 | 0.204 | |
| ID (Slope – Year-Scaled) | 0.011 | 0.104 | |
| Fixed Effect | Estimate | SE | z value |
| (Intercept) | 1.462 | 0.131 | 11.149 |
| Scaled Year | 0.079 | 0.051 | 1.545 |

**Appendix 2—table 9.** Intercept and slope for individual random effects for nut metrics of total time, total number of actions, and total number of strikes.

Values rounded to 3.d.p.

| Metric | Individual | Intercept | Year-Scaled (Slope) |
|---|---|---|---|
| | Fana | 2.614 | 0.064 |
| | Jire | 2.508 | 0.089 |
| | TUA | 2.328 | 0.036 |
| | Velu | 2.635 | 0.068 |
| Total Time | Yo | 3.200 | 0.128 |
| | Fana | 2.685 | –0.028 |
| | Jire | 2.507 | 0.049 |
| | TUA | 2.416 | 0.043 |
| | Velu | 2.399 | –0.050 |
| Actions | Yo | 2.814 | 0.098 |
| | Fana | 1.415 | 0.008 |
| | Jire | 1.321 | 0.051 |
| | TUA | 1.290 | 0.195 |
| | Velu | 1.476 | –0.043 |
| Strikes | Yo | 1.820 | 0.186 |

**Appendix 2—table 10.** Each unique action type employed by Yo in 1999 and 2016, described as a proportion of the first 1000 actions observed for Yo in each year.

The difference between years is calculated, and whether this difference is greater than 0.05 (5%) is determined. A direction of change between years is also indicated for each action.

| Action | 1999 | 2016 | 2016–1999 | Difference >0.05? | direction |
|---|---|---|---|---|---|
| brush SHELL | 0.01632302 | 0.00981354 | 0.00650948 | FALSE | decrease |
| drop ANVIL | 0.00429553 | 0.00392542 | 0.00037012 | FALSE | decrease |
| drop HAMMER | 0.03006873 | 0.00785083 | 0.02221789 | FALSE | decrease |
| drop KERNEL | 0.00085911 | 0.00294406 | 0.00208496 | FALSE | increase |
| drop NUT | 0.00429553 | 0.00392542 | 0.00037012 | FALSE | decrease |
| drop SHELL | 0.01546392 | 0.02551521 | 0.01005129 | FALSE | increase |
| eat KERNEL | 0.08676976 | 0.13837095 | 0.05160119 | TRUE | increase |

*Appendix 2—table 10 Continued on next page*

*Appendix 2—table 10 Continued*

| Action | 1999 | 2016 | 2016–1999 | Difference >0.05? | direction |
|---|---|---|---|---|---|
| flip ANVIL | 0.00257732 | 0 | 0.00257732 | FALSE | decrease |
| flip HAMMER | 0.00085911 | 0 | 0.00085911 | FALSE | decrease |
| flip KERNEL | 0.00085911 | 0 | 0.00085911 | FALSE | decrease |
| grasp ANVIL | 0.00429553 | 0.00392542 | 0.00037012 | FALSE | decrease |
| grasp HAMMER | 0.04553265 | 0.00588813 | 0.03964452 | FALSE | decrease |
| grasp KERNEL | 0.07044674 | 0.05888126 | 0.01156548 | FALSE | decrease |
| grasp NUT | 0.11340206 | 0.05691855 | 0.05648351 | TRUE | decrease |
| kiss KERNEL | 0.00257732 | 0 | 0.00257732 | FALSE | decrease |
| kiss NUT | 0.00085911 | 0 | 0.00085911 | FALSE | decrease |
| pass HAMMER | 0.00085911 | 0 | 0.00085911 | FALSE | decrease |
| pass KERNEL | 0.00171821 | 0 | 0.00171821 | FALSE | decrease |
| pass NUT | 0.01718213 | 0.00196271 | 0.01521942 | FALSE | decrease |
| peelhand SHELL | 0.00257732 | 0.00392542 | 0.0013481 | FALSE | increase |
| peelteeth SHELL | 0.05584192 | 0.12365064 | 0.06780871 | TRUE | increase |
| place HAMMER | 0.01202749 | 0.00196271 | 0.01006478 | FALSE | decrease |
| place KERNEL | 0.00601375 | 0.00098135 | 0.00503239 | FALSE | decrease |
| place NUT | 0.09621993 | 0.0451423 | 0.05107763 | TRUE | decrease |
| relocate | 0.00687285 | 0.00294406 | 0.00392879 | FALSE | decrease |
| reorient ANVIL | 0.00945017 | 0.00392542 | 0.00552475 | FALSE | decrease |
| rollhand HAMMER | 0.00085911 | 0 | 0.00085911 | FALSE | decrease |
| spit SHELL | 0.00085911 | 0.00098135 | 0.00012225 | FALSE | increase |
| strikeonehand HAMMER | 0.35395189 | 0.44455348 | 0.09060159 | TRUE | increase |
| strikeonehand NUT | 0.00085911 | 0 | 0.00085911 | FALSE | decrease |
| supportfoot ANVIL | 0.00343643 | 0.00294406 | 0.00049236 | FALSE | decrease |
| touchfoot ANVIL | 0.00085911 | 0 | 0.00085911 | FALSE | decrease |
| touchhand ANVIL | 0.00429553 | 0.00392542 | 0.00037012 | FALSE | decrease |
| touchhand HAMMER | 0.00515464 | 0 | 0.00515464 | FALSE | decrease |
| touchhand KERNEL | 0.00601375 | 0.00294406 | 0.00306968 | FALSE | decrease |
| touchhand NUT | 0.01460481 | 0.03140334 | 0.01679853 | FALSE | increase |
| touchhand SHELL | 0.00085911 | 0.00392542 | 0.00306631 | FALSE | increase |
| bite SHELL | 0 | 0.00098135 | 0.00098135 | FALSE | increase |
| brush HAMMER | 0 | 0.00098135 | 0.00098135 | FALSE | increase |
| grasp SHELL | 0 | 0.00294406 | 0.00294406 | FALSE | increase |
| reorient HAMMER | 0 | 0.00098135 | 0.00098135 | FALSE | increase |
| supporthand ANVIL | 0 | 0.00098135 | 0.00098135 | FALSE | increase |

## Appendix 3

### Coula Nut Cracking

Results

Three of our focal individuals were observed cracking coula nuts in 2011 (Jire, Tua & Yo). Coula nuts do not occur naturally at Bossou, but are cracked by chimpanzees at other sites across Africa (*Boesch and Boesch, 1983*; *Biro et al., 2006*) and have historically been experimentally provided to individuals at Bossou during selected field seasons (*Biro et al., 2003*; *Biro et al., 2006*). Of the three individuals who were observed cracking coula nuts in 2011, only Jire was observed also cracking oil-palm nuts in the same year. Coula nuts require somewhat more effort to crack and process than smaller oil-palm nuts (*Boesch and Boesch, 1983*); however, for two individuals (Jire and Tua), coula nuts were cracked and processed with comparable efficiency to smaller oil-palm nuts (with a slight increase in total time taken, the number of actions and unique action types used, and the total number of strikes of the hammer stone; see *Appendix 3—table 1* & *Figure 1* for corresponding data). On average, Jire took an extra 22.8 s to crack coula nuts compared with oil-palm nuts in 2011. Similarly, Tua took an additional 44.1 s to crack coula nuts in 2011, compared with oil-palm nuts in 2008 (the closest comparison point for Tua, who was not observed cracking oil-palm nuts in 2011).

Conversely, Yo showed a considerably lower efficiency cracking coula nuts in 2011 compared with her efficiency cracking oil-palm nuts in the previously sampled field season (2008; see *Appendix 3— figure 1*). On average, Yo took an additional 2.63 minutes (+157.9 s; mean duration across oil-palm nuts in 2008 and coula nuts in 2011) to crack open coula nuts, during which Yo performed an average of 78 additional actions to crack the nut and consume all associated kernel (including an additional 23 hammer strikes; measured as using median values for each nut species). For Yo, coula nuts frequently rolled off of the anvil stone, resulting in Yo placing each coula nut on the anvil 11 times prior to successful cracking (median value; in 2008, Yo would place oil-palm nuts on the anvil once per nut). Additionally, Yo exhibited exceptional variability between the number of times she switched out tools during coula-nut cracking (interquartile range = 7.25, for oil-palm nuts in 2008 this was 0), as well as for the number of tool adjustments per coula nut (coula = 4.25, oil-palm=0). Overall, these metrics suggest that Yo found coula-nut cracking in 2011 disproportionately difficult compared with oil-palm nut cracking. This was to an extent which was not mirrored by other individuals of similar ages, as well as compared to her own performance when cracking oil-palm nuts in previous and subsequent years (see *Appendix 3—table 1*, and *Video 2*).

### Discussion

The disproportionate difficulty that Yo faced cracking coula nuts was surprising given the history of this behavior at Bossou (*Matsuzawa, 1994*; *Biro et al., 2003*; *Biro et al., 2006*). Unlike oil-palm nuts, whose trees occur within the home range of the Bossou community, coula nuts are not found at Bossou (*Biro et al., 2003*; *Biro et al., 2006*). Coula nuts were first introduced to chimpanzees at the outdoor laboratory in 1993, when Yo was the only adult individual to begin cracking coula nuts without engaging in a phase of exploratory behavior (*Biro et al., 2006*). Other individuals began cracking coula nuts through successive, intermittent presentations of coula nuts across further field seasons (1996, 2000, 2002, 2005). Given that Yo readily engaged in coula nut cracking, it was deduced that Yo was likely an immigrant from the neighboring Yealé population, where both oil-palm and coula nuts are readily available (*Biro et al., 2003*; *Biro et al., 2006*). Therefore, Yo likely has the longest history and greatest experience cracking coula nuts, despite exhibiting such low efficiency in 2011.

There are numerous hypotheses for why Yo showed a dramatic reduction in efficiency when cracking coula nuts in 2011. It is important to note that, for Yo, data for the cracking of coula nuts in this year came from a single, three-hour long encounter, where Yo spent one hour and 17 min of continuous effort cracking the first 20 coula nuts. It is possible that a short-term effect (such as dehydration, hunger, physical injury) may have inhibited the cognitive processes or physical movements required for streamlined tool action. However, we noticed no obvious signs of physical injury, and Yo spent the majority of the encounter prioritizing nut cracking over both the water point and the more readily available oil-palm fruits. This reduces the likelihood of these hypotheses explaining the observed changes in Yo's behavior. Alternatively, it may be that the reduced ability to crack coula nuts in 2011 was a product of senescence, a hypothesis which is supported by the fact that Yo also exhibited the largest reduction in efficiency when cracking oil-palm nuts at older ages

(which continued to intensify further in later life, past the 2016 field season; see below). Given that coula nuts are larger and rounder than oil-palm nuts (and thus can be more liable to roll off of anvil stones) and have more fibrous outer shells (*Boesch and Boesch, 1983*), they therefore may require heavier hammer stones, and/or more forceful or elaborate sequences of striking and peeling actions to separate shell from kernel. Physiological senescence – such as reduced strength of dexterity - may have rendered coula nut cracking even more challenging than the cracking of oil-palm nuts, for which Yo was also exhibiting increasing difficulty. Alternatively, cognitive senescence may have also contributed to the additional difficulty Yo faced when cracking coula nuts, where she may have struggled to generate suitable action patterns to crack open a less frequently encountered nut species.

Given the absence of data for coula nut cracking in earlier years, we emphasize that these conclusions are tentative, and further data collection is required for coula-nut cracking over different field seasons.

**Appendix 3—table 1.** Summary of nut cracking and processing metrics for Tua, Jire, and Yo between 2008 and 2016.

Total time duration is summarized by mean and standard deviation. All other metrics are summarized by medians and interquartile ranges. Horizontal lines indicate where no data is available for a given year. Information on how each metric is defined and estimated can be found in the Methods.

| Individual | Metric | Year and Nut Type | | | |
|---|---|---|---|---|---|
| | | 2008 | 2011 | | 2016 |
| TUA | | oil-palm (n=81) | coula (n=26) | | - |
| | Means (sd) | | | | |
| | Total Time (s) | 10.6 (6.34) | 54.7 (92.6) | | - |
| | Medians (IQR) | | | | |
| | Actions | 8 (4) | 25.5 (16) | | - |
| | Action Types | 6 (2) | 8 (3) | | - |
| | Strikes | 2 (2) | 7 (6.75) | | - |
| | Nut Positioning | 1 (0) | 2 (2.75) | | - |
| | Tool Reorientations | 0 (0) | 0 (0.75) | | - |
| | Tool Changes | 0 (0) | 0 (0) | | - |
| Jire | | oil-palm (n=104) | oil-palm (n=25) | coula (n=17) | oil-palm (n=82) |
| | Means (sd) | | | | |
| | Total Time (s) | 12.8 (8.37) | 21.8 (14.9) | 44.6 (21.6) | 21.9 (22.3) |
| | Medians (IQR) | | | | |
| | Actions | 8 (5) | 11 (6) | 26 (17) | 11 (7.75) |
| | Action Types | 6 (2) | 7 (3) | 11 (2) | 7 (2) |
| | Strikes | 2 (2) | 3 (3) | 7 (4) | 4 (2) |
| | Nut Positioning | 1 (1) | 1 (1) | 3 (1) | 1 (1) |
| | Tool Reorientations | 0 (0) | 0 (0) | 0 (1) | 0 (0) |
| | Tool Changes | 0 (0) | 0 (0) | 0 (0) | 0 (0) |
| Yo | | oil-palm (n=141) | coula (n=20) | | oil-palm (n=33) |
| | Means (sd) | | | | |
| | Total Time (s) | 28.1 (24.6) | 186 (183.4) | | 55.9 (42.3) |

*Appendix 3—table 1 Continued on next page*

*Appendix 3—table 1 Continued*

| Individual | Metric | Year and Nut Type | | |
| --- | --- | --- | --- | --- |
| | | **2008** | **2011** | **2016** |
| | *Medians (IQR)* | | | |
| | Actions | 11 (6) | 88.5 (60.5) | 27 (14) |
| | Action Types | 7 (1) | 11.5 (8.5) | 8 (1) |
| | Strikes | 4 (2) | 27 (23) | 12 (8) |
| | Nut Positioning | 1 (1) | 11 (22.25) | 2 (1) |
| | Tool Reorientations | 0 (0) | 0.5 (4.25) | 0 (0) |
| | Tool Changes | 0 (0) | 0 (7.25) | 0 (0) |

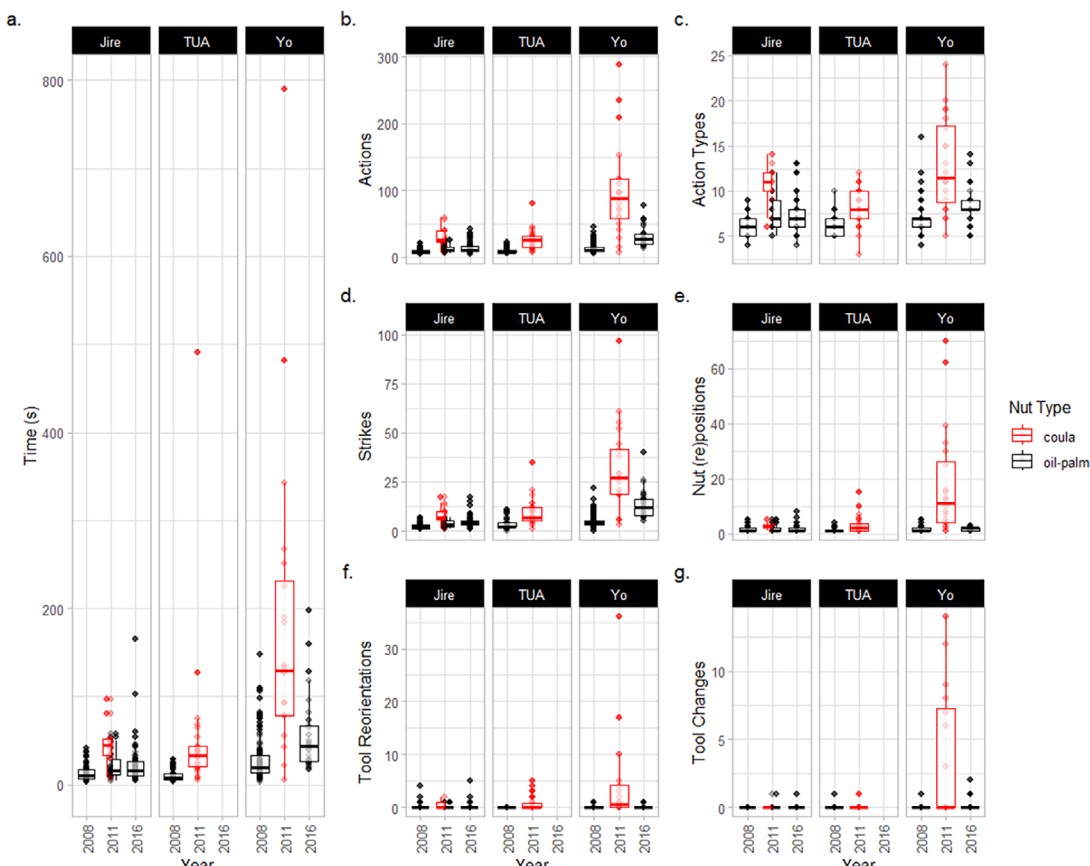

**Appendix 3—figure 1.** Metrics for the cracking and processing of both oil-palm and coula nuts. Data is confined to individuals who cracked nuts from both species. Data for coula nuts is in red, and data for oil-palm nuts is in black. Data describes the cracking and processing of individual nuts, including (**a**) the total time taken; (**b**) the total number of actions used; (**c**) the number of unique types of actions used; (**d**) the number of hammer strikes; (**e**) the number of times the nut had to be placed and replaced on the anvil; (**f**) the number of reorientations of stone tools, and (**g**) the number of times stone tools were switched over, or switched out for new tools.

## Appendix 4

### Oil-palm nut cracking for Yo in 2018

Data collection at the outdoor laboratory (second location) was conducted in 2018, offering an additional year for our analysis. We did not, however, include this data in our main manuscript for several reasons.

Data was collected using camera traps in 2018, which were motion-triggered. These camera traps were only able to capture a limited viewpoint of the outdoor laboratory and did not record encounters continuously. Therefore, estimating the entire party composition during encounters, the behaviors of individuals as they moved around the outdoor laboratory, and the duration of stone-tool selection events were not possible, as individuals were often out of sight for some or all of the behavior. We also could not collect fine-grained action sequence data for the entire behavioral sequence used to crack individual nuts, as there were gaps in the video footage lasting several seconds. Additionally, in 2018, only three elderly individuals remained at Bossou (Yo, Jire, and Fana), and the majority of clear video footage concerning nut cracking was of Yo. All nuts cracked by Yo in this year were oil-palm nuts.

Whilst the collection of fine-grained action sequence data was not possible, it was possible to record the start and end times of nut cracking sequences aimed at individual nuts using timestamps on video footage. We therefore collected the start and end times of nut cracking sequences performed by Yo. We filtered our data to only include nuts where we could clearly identify the start and end of Yo's nut cracking behaviors across videos, using the same coding scheme reported in the methods of our manuscript (see *Appendix 4—table 1*).

To calculate a mean duration of oil-palm nut cracking for Yo in 2018, we first found the mean within each encounter, and then calculated a mean across encounters. This blocking was used to compensate for the non-independence of nuts cracked as part of the same encounter. Overall, it took Yo on average 74.2 s to crack each oil palm nut in 2018 (SD = 9.1 s; estimated using the means of each encounter). This was 18.3 s longer than the mean time taken in 2016, meaning that Yo became even less efficient at oil-palm nut cracking in later years. Overall, when comparing the mean duration of oil-palm nut cracking between 1999 and 2018, Yo experienced a +171% change in total time taken (1999=27.4 s; difference between years = +46.7 s).

**Appendix 4—table 1.** Total time taken for Yo to crack oil-palm nuts in 2018.
Encounter numbers indicate where data was collected for the same encounter. Nut number indicates which nut (in chronological order) Yo was cracking for a specific encounter. Numbers missing from this chronology were removed as we could not reliably estimate start and end times.

| Encounter | Nut Number | Duration (s) |
|---|---|---|
| 9 | 6 | 145 |
| 9 | 7 | 73 |
| 9 | 8 | 24 |
| 9 | 9 | 29 |
| 11 | 1 | 220 |
| 11 | 3 | 173 |
| 11 | 4 | 165 |
| 11 | 5 | 54 |
| 11 | 6 | 42 |
| 11 | 7 | 17 |
| 11 | 8 | 36 |
| 11 | 9 | 75 |
| 11 | 10 | 37 |

*Appendix 4—table 1 Continued on next page*

*Appendix 4—table 1 Continued*

| Encounter | Nut Number | Duration (s) |
|---|---|---|
| 11 | 17 | 35 |
| 11 | 18 | 59 |
| 11 | 19 | 54 |

