## [Editor Report · eLife Assessment]

This **valuable** study provides a novel framework for leveraging longitudinal field observations to examine the effects of aging on stone tool use behaviour in wild chimpanzees. The methods and results are robust providing **solid** evidence of the effects of old age on nut cracking behaviour at this field site. Despite the low sample size of five individuals, this study is of broad interest to ethologists, primatologists, archaeologists, and psychologists.

---

## [Referee Report · Reviewer #1 (Public review)]

Summary:

Howard-Spink et al. investigated how older chimpanzees changed their behavior regarding stone tool use for nutcracking over a period of 17 years, from late adulthood to old age. This behavior is cognitively demanding, and it is a good target for understanding aging in wild primates. They used several factors to follow the aging process of five individuals, from attendance at the nut-cracking outdoor laboratory site to time to select tools and efficiency in nut-cracking to check if older chimpanzee changed their behavior.

Indeed, older chimpanzees reduced their visits to the outdoor lab, which was not observed in the younger adults. The authors discuss several reasons for that; the main ones being physiological changes, cognitive and physical constraints, and changes in social associations. Much of the discussion is hypothetical, but a good starting point, as there is not much information about senescence in wild chimpanzees.

The efficiency for nut-cracking was variable, with some individuals taking a long time to crack nuts while others showed little variance. As this is not compared with the younger individuals and the sample is small (only five individuals), it is difficult to be sure if this is also partly a normal variance caused by other factors (ecology) or is only related to senescence.

Strengths:

(1) 17 years of longitudinal data in the same setting, following the same individuals.

(2) Using stone tool use, a cognitively demanding behavior, to understand the aging process.

Weaknesses:

A lack of comparison of the stone tool use behavior with younger individuals in the same period, to check if the changes observed are only related to age or if it is an overall variance. The comparison with younger chimpanzees was only done for one of the variables (attendance).

Comments on Revised Version (from BRE):

The authors have now added to the manuscript that they did not have sufficient data to compare additional variables to younger chimpanzees, and therefore compared intra-individual variation across field seasons. They have also explained that nut hardness, although not measured, was largely controlled for due to the experimental nature of the 'outdoor laboratory' whereby only nuts of a suitable maturity (and hardness) are provided to the chimpanzees. The discussion now also includes mention of other ecological variables and their potential influence on the results.

---

## [Referee Report · Reviewer #2 (Public review)]

Summary:

Primates are a particularly important and oft-applied model for understanding the evolution of, e.g., life history and senescence in humans. Although there is a growing body of work on aging in primates, there are three components of primate senescence research that have been underutilized or understudied: (1) longitudinal datasets, (2) wild populations, and (3) (stone) tool-use behaviors. Therefore, the goal of this study was to (1) use a 17-year longitudinal dataset (2) of wild chimpanzees in the Bossou forest, (3) visiting a site for field experiments on nut-cracking. They sampled and analyzed data from five field seasons for five chimpanzees of old age. From this sample, Howard-Spink and colleagues noted a decline in tool-use and tool-use efficiency in some individuals, but not in others. The authors then conclude that there is a measurable effect of senescence on chimpanzee behavior, but that it varies individually. The study has major intellectual value as a building block for future research, but there are several major caveats.

Strengths:

With this study, Howard-Spink and colleagues make a foray into a neglected topic of research: the impact of the physiological and cognitive changes due to senescence on stone tool use in chimpanzees. Based on novelty alone, this is a valuable study. The authors cleverly make use of a longitudinal record covering 17 years of field data, which provides a window into long-term changes in the behavior of wild chimpanzees, which I agree cannot be understood through cross-sectional comparisons.

The metrics of 'efficiency' (see caveats below) are suitable for measuring changes in technological behavior over time, as specifically tailored to the nut-cracking (e.g., time, number of actions, number of strikes, tool changes). The ethogram and the coding protocol are also suitable for studying the target questions and objectives. I would recommend, however, the inclusion of further variables that will assist in improving the amount of valid data that can be extrapolated (see also below).

With this pilot, Howard-Spink and colleagues have established a foundation upon which future research can be designed, including further investigation with the Bossou dataset and other existing video archives, but especially future targeted data collection, which can be designed to overcome some of the limits and confounds that can be identified in the current study.

Weaknesses:

Although I agree with the reasoning behind conducting this research and understand that, as the authors state, there are logistical considerations that have to be made when planning and executing such a study, there are a number of methodological and theoretical shortcomings that either need to be more explicitly stated by the authors or would require additional data collection and analysis.

One of the main limitations of this study is the small sample size. There are only 5 of the old-aged individuals, which is not enough to draw any inferences about aging for chimpanzees more generally. Howard-Spink and colleagues also study data from only five of the 17 years of recorded data at Bossou. The selection of this subset of data requires clarification: why were these intervals chosen, why this number of data points, and how do we know that it provides a representative picture of the age-related changes of the full 17 years?

With measuring and interpreting the 'efficiency' of behaviors, there are in-built assumptions about the goals of the agents and how we can define efficiency. First, it may be that efficiency is not an intentional goal for nut-cracking at all, but rather, e.g., productivity as far as the number of uncrushed kernels (cf. Putt 2015). Second, what is 'efficient' for the human observer might not be efficient for the chimpanzee who is performing the behavior. More instances of tool-switching may be considered inefficient, but it might also be a valid strategy for extracting more from the nuts, etc. Understanding the goals of chimpanzees may be a difficult proposition, but these are uncertainties that must be kept in mind when interpreting and discussing 'decline' or any change in technological behaviors over time.

For the study of the physiological impact of senescence of tool use (i.e., on strength and coordination), the study would benefit from the inclusion of variables like grip type and (approximate) stone size (Neufuss et al., 2016). The size and shape of stones for nut-cracking have been shown to influence the efficacy and 'efficiency' of tool use (i.e., the same metrics of 'efficiency' implemented by Howard-Spink et al. in the current study), meaning raw material properties are a potential confound that the authors have not evaluated.

Similarly, inter- and intraspecific variation in the properties of nuts being processed is another confound (Falótico et al., 2022; Proffitt et al., 2022). If oil palm nuts were varying year-to-year, for example, this would theoretically have an effect on the behavioral forms and strategies employed by the chimpanzees, and thus, any metric of efficiency being collected and analyzed. Further, it is perplexing that the authors analyze only one year where the coula nuts were provided at the test site, but these were provided during multiple field seasons. It would be more useful to compare data from a similar number of field seasons with both species if we are to study age-related changes in nut processing over time (one season of coula nut-cracking certainly does not achieve this).

Both individual personality (especially neophilia versus neophobia; e.g., Forss & Willems, 2022) and motivation factors (Tennie & Call, 2023) are further confounds that can contribute to a more valid interpretation of the patterns found. To draw any conclusions about age-related changes in diet and food preferences, we would need to have data on the overall food intake/preferences of the individuals and the food availability in the home range. The authors refer briefly to this limitation, but the implications for the interpretation of the data are not sufficiently underlined (e.g., for the relevance of age-related decline in stone tool-use ability for individual survival).

Generally speaking, there is a lack of consideration for temporal variation in ecological factors. As a control for these, Howard-Spink and colleagues have examined behavioral data for younger individuals from Bossou in the same years, to ostensibly show that patterns in older adults are different from patterns in younger adults, which is fair given the available data. Nonetheless, they seem to focus mostly on the start and end points and not patterns that occur in between. For example, there is a curious drop in attendance rate for all individuals in the 2008 season, the implications of which are not discussed by the authors.

As far as attendance, Howard-Spink and colleagues also discuss how this might be explained by changes in social standing in later life (i.e., chimpanzees move to the fringes of the social network and become less likely to visit gathering sites). This is not senescence in the sense of physiological and cognitive decline with older age. Instead, the reduced attendance due to changes in social standing seems rather to exacerbate signs of aging rather than be an indicator of it itself. The authors also mention a flu-like epidemic that caused the death of 5 individuals; the subsequent population decline and related changes in demography also warrant more discussion and characterization in the manuscript.

Understandably, some of these issues cannot be evaluated or corrected with the presented dataset. Nonetheless, these undermine how certain and/or deterministic their conclusions can really be considered. Howard-Spink et al. have not strongly 'demonstrated' the validity of relationships between the variables of the study. If anything, their cursory observations provide us with methods to apply and hypotheses to test in future studies. It is likely that with higher-resolution datasets, the individual variability in age-related decline in tool-use abilities will be replicated. For now, this can be considered a starting point, which will hopefully inspire future attempts to research these questions.

Falótico, T., Valença, T., Verderane, M. & Fogaça, M. D. Stone tools differences across three capuchin monkey populations: food's physical properties, ecology, and culture. Sci. Rep. 12, 14365 (2022).

Forss, S. & Willems, E. The curious case of great ape curiosity and how it is shaped by sociality. Ethology 128, 552-563 (2022).

Neufuss, J., Humle, T., Cremaschi, A. & Kivell, T. L. Nut-cracking behaviour in wild-born, rehabilitated bonobos (*Pan paniscus*): a comprehensive study of hand-preference, hand grips and efficiency. Am. J. Primatol. 79, e22589 (2016).

Proffitt, T., Reeves, J. S., Pacome, S. S. & Luncz, L. V. Identifying functional and regional differences in chimpanzee stone tool technology. R. Soc. Open Sci. 9, 220826 (2022).

Putt, S. S. The origins of stone tool reduction and the transition to knapping: An experimental approach. J. Archaeol. Sci.: Rep. 2, 51-60 (2015).

Tennie, C. & Call, J. Unmotivated subjects cannot provide interpretable data and tasks with sensitive learning periods require appropriately aged subjects: A Commentary on Koops et al. (2022) "Field experiments find no evidence that chimpanzee nut cracking can be independently innovated". ABC 10, 89-94 (2023).

Comments on Revised Version (from BRE):

The authors have revised their methods to clarify why certain field seasons were chosen and have clarified aspects of their analysis relevant to this reviewer's concerns. The coula nut cracking data and results which were of a single season have now been restricted to the Supplementary. The revised discussion now includes a much more detailed limitations section including both ecological factors but also the effects of social aging. Stone tool size, grip and other factors are also acknowledged as being potentially important for measuring efficiency but the authors were unable to include in this study due to the nature of the dataset.

---

## [Author Response]

The following is the authors’ response to the original reviews

The main criticisms levied by both reviewers can be traced down to our use of a long-term video archive to assess for the effects of aging on individual chimpanzees over extended time periods. Specifically, the reviewers raised several points surrounding whether we could exclude ecological variation over years as the explanation of changes with aging, rather than aging itself. Whilst we acknowledge there are limitations to our approach, we provide a comprehensive response to these points highlighting:

(1) Where ecological variables have been accounted for using controls (including the behaviors of other individuals, or an aging individuals’ behavior at younger ages).

(2) Where ecological data may be missing, thus a potential limitation to our study, and further data would be beneficial.

(3) Whether, in light of these limitations, interannual ecological variation offers a likely explanation for the behavioral changes we have identified. We provide an argument that whilst ecological data would be desirable for our study, interannual changes in ecology are unlikely to explain the trends in our data. Additionally, we explain why age-related changes, such as senescence, are more likely to underpin the patterns described in our manuscript.

Across 1-3, we have made substantial changes to the reporting of our manuscript to ensure that our results are communicated transparently, and conclusions are made with appropriate care. We have also moved all discussion of coula-nut cracking to the supplementary materials, given the points raised by reviewers about the lack of data describing coula-nut cracking in earlier field seasons.

We hope that these modifications will enhance both the editors’ and reviewers’ assessment of our manuscript, where we have aimed to make careful conclusions that are supported by our available data. Similarly, we have aimed to communicate the importance of our results across fields of research including primatology, evolutionary anthropology, and comparative gerontology, and hope that our research will be of use to further studies within these subfields.

**Reviewer 1 (Recommendations for the authors):**
(1) If possible, include results or a summary of the behaviour of younger adults using stone tools during the same period. It would be helpful to know if they had the same or different pattern to exclude other factors that may influence the tool use (harder nuts in a particular season, diseases, motivation for other foods, etc).

We include data for other individuals when analyzing attendance. However, we did not collect comparable long-term efficiency data on younger adult individuals for this study. This is, in part, due to the time constraints imposed by long-term behavior coding. Additionally, only one adult was both present at Bossou throughout the 1999-2016 period, and younger than the threshold for our old-age category across these years (thus, the baseline used to compare with older adults would be just one younger adult, thus would not have been useful for characterizing normal variation of many younger adults over time). However, given the longitudinal data we present, we can use data from the earlier field seasons for each elderly focal individual as a personalized baseline control. Previous studies at Bossou find that across the majority of adulthood, efficiency varies between individuals, but is stable within individuals over time (e.g., Berdugo et al. 2024, cited). We detected similar stability in individuals’ efficiency over the first three field seasons sampled in our analysis, where there was very little intra-individual variation in tool-using efficiency. However, in later years, two individuals (Velu & Yo) began to exhibit relatively large reductions in efficiency.

These results are unlikely to be explained by ecological variation. If there was a change in ecology underpinning our results, we would expect: [1] changes in ecology to also introduce variation in earlier field seasons, and [2] to influence all individuals in our study similarly. As such, if the changes observed in later field seasons were due to ecological changes, they should have caused a reduced efficiency across individuals, and to a similar degree – we did not observe this result, with large reductions in efficiency were confined to two individuals. Moreover, for Yo (the individual who exhibited the largest reduction in efficiency) we found some additional evidence that changes in oil-palm-nut cracking efficiency extended beyond the period we sampled, i.e. they were evident even in 2018, reflecting a long-term, directional reduction in efficiency as compared to earlier years of her life. This consistent reduction in tool-using efficiency over multiple years adds further weight to the hypothesis that changes at the level of the individual were causing reduced tool-using efficiency, rather than our results being underpinned by interseasonal variation in ecology.

Whilst we agree that our study is limited in the extent to which we can analytically assess ecological explanations for changes in nut-cracking efficiency, we believe that hypothetical ecological changes across field seasons do not predict our results. We now raise both sides of this debate in our discussion, where we outline our limitations (see lines 535-593).

(2) The data from 2011 was scarce, with only one individual having 10 encounters. It would be better to be cautious with this season's results.

We appreciate this limitation raised by the reviewer. Velu and Yo were only encountered a few times in 2011; however, both were encountered more frequently in 2016. For 2011, we did not collect oil-palm nut cracking data for either Yo or Velu. Thus, their change in efficiency was detected by models using data from all other years, regardless of the few encounters in 2011. This sparsity of data may still have influenced our metrics for the proportion of time chimpanzees spent engaging in different behaviors when present at the outdoor laboratory in 2011, particularly for Velu, who was one of the two individuals who exhibited a change in behavior in this year (along with Fana, N = 10 for 2011). We have therefore added a line in our results and discussion highlighting the sparsity of data for Velu when estimating these proportions for 2011 (see lines 255-256 & 410).

Minor corrections(1) The last paragraph of the introduction presents many results, which should be in the results section.

We would like to keep this section of the introduction. Our paper investigates the effect of aging on many different aspects of nut cracking, which could become confusing for readers unless laid out clearly. We believe that having a short summary early on in the paper assists readers with following the methods and arguments presented within our paper.

(2) The first section (Sampled data) of the results contains much information that belongs in the methods section.

We appreciate that there is some overlap between our methods and results section. However as the results section comes before the methods in our manuscript, we wanted to ensure that there is suitable information in our results that allow our results to be interpreted clearly by readers, and that the methods used to generate these results are transparently communicated. For these reasons, we will leave this information in the results, as we believe it increases our paper’s readability.

**Reviewer 2 (Public review):**
One of the main limitations of this study is the small sample size. There are only 5 of the old-aged individuals, which is not enough to draw any inferences about aging for chimpanzees more generally. Howard-Spink and colleagues also study data from only five of the 17 years of recorded data at Bossou. The selection of this subset of data requires clarification: why were these intervals chosen, why this number of data points, and how do we know that it provides a representative picture of the age-related changes of the full 17 years?

We note that our sample size is limited to 5 individuals. This is an inevitable constraint of analyzing aging longitudinally in long-lived species, as only few individuals will live to old age. We argue that 17 years is a long enough period of study, as in the initially sampled field season (1999) focal individuals are reaching a mature age of adulthood (39-44 years) and begin to age progressively up to ages that are typically considered to be on the extreme side for chimpanzees’ lifespans in the wild (56-61 years). We raise in our methods that whilst it is difficult to determine precisely when chimpanzees become ‘old aged’, previous studies use the age of around 40 years, as from this age survivorship begins to decrease more rapidly (see Wood et al., Science 2023). Indeed, one focal individual (Tua) disappeared during the period of our study (presumed dead), and one other individual died in 2017 (Velu), the year after our final sampled field season. As of 2025, two other focal females have since died, and only one focal individual was still alive at Bossou (Jire, the individual exhibiting the least evidence for senescence over our study period). These observations suggest that we successfully captured data from chimpanzees during the oldest ages of their lives for most individuals in the community. Moreover, the period of 1999-2016 contains the majority of data available within the Bossou Archive, with years before and after this window containing comparably less data. This information is included within our results and methods (see sections 2.1 and 4.1).

For our earliest field season (1999), it is unlikely that senescence had already had an effect on stone-tool use, as we measured efficiency to be high across all efficiency metrics for all individuals. For example, in 1999, the median number of hammer strikes performed by focal chimpanzees ranged from 2-4 strikes, and this was comparable to the efficiency reported across all adults observed in previous studies at Bossou (Biro et al. 2003, Anim. Cog.). This finding suggests that senescence effects had not yet taken place, allowing us to evaluate whether aging affects efficiency over subsequent field seasons. This point is now included in the manuscript on lines 449-452.

We sampled at 4-to-5-year intervals to balance the time-intensive nature of fine-scale behavior coding against the need to sample data across the extended 17-year time window available in our study. We limited the final year to 2016 as, in following years, data were collected using different sampling protocols (though, see limited data from 2018 in the supplementary materials). We aimed to keep the intervals between years as consistent as possible (approx. 4 years); however, for some years data were not collected at Bossou, due to disease outbreaks in the region. In these instances, we selected the closest field season where suitable data were available for study (always +/- 1 year). We have provided further clarification surrounding our sampling regime in the methods (see amendments in section 4.1)

With measuring and interpreting the 'efficiency' of behaviors, there are in-built assumptions about the goals of the agents and how we can define efficiency. First, it may be that efficiency is not an intentional goal for nut-cracking at all, but rather, e.g., productivity as far as the number of uncrushed kernels (cf. Putt 2015). Second, what is 'efficient' for the human observer might not be efficient for the chimpanzee who is performing the behavior. More instances of tool-switching may be considered inefficient, but it might also be a valid strategy for extracting more from the nuts, etc. Understanding the goals of chimpanzees may be a difficult proposition, but these are uncertainties that must be kept in mind when interpreting and discussing 'decline' or any change in technological behaviors over time.

We agree that knowing precisely how chimpanzees perceive their own efficiency during tool use is unlikely to be available through observation alone. However, under optimal foraging theory, it is reasonable to assume that animals aim to economize foraging behaviors such that they maximize their rate of energy intake. Moreover, a wealth of studies demonstrate that adult chimpanzees acquire and refine tool-using skill efficiency throughout their lives. For example, during nut cracking, adults often select tools with specific properties that aid efficient nut cracking (Braun et al. 2025, J. Hum. Evol.; Carvalho et al. 2008, J. Hum. Evol.; Sirianni et al. 2015, Anim. Behav.); perform nut cracking using more streamlined combinations of actions than less experienced individuals (Howard-Spink et al. 2024, Peer J; Inoue-Nakamura & Matsuzawa 1997, J. Comp. Psychol.), and as a result end up cracking nuts using fewer hammer strikes, indicating a higher level of skill (Biro et al. 2003, Anim. Cogn.; Boesch et al. 2019, Sci. Rep.). Ultimately, these factors suggest that across adulthood, experienced chimpanzees perform nut cracking with a level of efficiency which exceeds novice individuals, including across the whole behavioral sequence for tool use, even if they are not aware or intending to do so. Previous studies at Bossou have also highlighted that there are stable inter-individual differences in efficiency of individuals over time (Berdugo et al. 2024, Nat. Hum. Behav.). This pattern of findings allows us to ask whether this acquired level of skill is stable across the oldest years of an individual’s life, or whether some individuals experience decreased efficiency with age. In addition, our selection of efficiency metrics is in keeping with a wealth of studies which examine the efficiency of stone-tool using in apes, thus, we argue that this is not problematic for our study.

As we stated in our initial responses to reviewers, it is unlikely that tool switching is a valid strategy for tool use, as it is so rarely performed by proficient adult nut crackers (including earlier in life for our focal individuals). Nevertheless, we did not find a significant change in tool switching for oil-palm nut cracking, and this behavioral change was only observed when Yo was cracking coula nuts. As we have now moved discussion of coula nut cracking to the supplementary materials (and tempered discussion of coula nut cracking to emphasize the need for more data) this behavioral variable does not influence our reported results.

In our discussion, we also highlight how seemingly less efficient actions may reflect a valid strategy for nut cracking. E.g. a greater number of tool strikes may reflect a strategy of compensation for progressive tool wear. This would still reflect a reduced efficiency (e.g. in terms of the rate at which kernels can be consumed), but may perhaps borne for the necessity to accommodate for changes in an individuals’ physical affordances with aging. Thus, we do take the Reviewer’s point into account, but by using an alternative, more likely, example given the available data. We have now emphasized this point in lines 521-527.

We have also clarified these matters by adding more information into our methods (see lines 798-802 and 828-829), highlighting that we take a perspective on efficiency that reflects the speed of nut processing and kernel consumption, and the number of different behavioral elements required to do so. Our phrasing now explicitly avoids using language that assumes that individuals’ have some perception of their own efficiency during tool use.

For the study of the physiological impact of senescence of tool use (i.e., on strength and coordination), the study would benefit from the inclusion of variables like grip type and (approximate) stone size (Neufuss et al., 2016). The size and shape of stones for nut-cracking have been shown to influence the efficacy and 'efficiency' of tool use (i.e., the same metrics of 'efficiency' implemented by Howard-Spink et al. in the current study), meaning raw material properties are a potential confound that the authors have not evaluated.

We did not collect this data as part of our study. Whilst grip type could be a useful variable to measure for future studies, it is not necessary to demonstrate senescence per se. However, we agree that this could be a fruitful avenue to understand changes in behavior at greater granularity, and have added this as a recommendation for further study. We also now provide a discussion on stone dimensions and materials as part of our limitations (see lines 581-589 for both points).

Similarly, inter- and intraspecific variation in the properties of nuts being processed is another confound (Falótico et al., 2022; Proffitt et al., 2022;). If oil palm nuts were varying year-to-year, for example, this would theoretically have an effect on the behavioral forms and strategies employed by the chimpanzees, and thus, any metric of efficiency being collected and analyzed. Further, it is perplexing that the authors analyze only one year where the coula nuts were provided at the test site, but these were provided during multiple field seasons. It would be more useful to compare data from a similar number of field seasons with both species if we are to study age-related changes in nut processing over time (one season of coula nut-cracking certainly does not achieve this).

We have moved all discussion of coula nuts to the supplementary materials so as to avoid any confusion with oil-palm nuts (see comments from Reviewer 2, and our response). Nut hardness may influence the difficulty with which nuts are cracked, with one of the most likely factors influencing nut hardness being its age: young nuts are relatively harder to crack, whereas older nuts, which are often worm-eaten or can be empty, crack more easily, yet are not worth cracking (Sakura & Matsuzawa, 1991; Ethology). We largely controlled for this in our study, as the nuts provided at outdoor laboratories were inspected to ensure that the majority of them were of suitable maturity for cracking, and we now clarify this control in our methods (see lines 678-680) and when discussing our study limitations (see lines 551-558). In these sections, we also highlight a previous study at Bossou that shows chimpanzees select nuts which can be readily cracked, based on their age (Sakura & Matsuzawa, 1991; Ethology).

We acknowledge that we are limited in the extent to which we can control for interannual variation in ecology with our available data. However, we highlight why interannual variability is unlikely to fully explain our results (see lines 551-580 and response to comments from Reviewer 1). We also highlight in our limitations section that future studies should (where possible) aim to collect more ecological data to account for possible confounds more rigorously.

Both individual personality (especially neophilia versus neophobia; e.g., Forss & Willems, 2022) and motivation factors (Tennie & Call, 2023) are further confounds that can contribute to a more valid interpretation of the patterns found. To draw any conclusions about age-related changes in diet and food preferences, we would need to have data on the overall food intake/preferences of the individuals and the food availability in the home range. The authors refer briefly to this limitation, but the implications for the interpretation of the data are not sufficiently underlined (e.g., for the relevance of age-related decline in stone tool-use ability for individual survival).

In our discussion, we highlight that multiple aging factors may influence apes’ dietary preferences and motivations to attend experimental (and perhaps also naturally-occurring) nut cracking sites (see lines 397-443 and 542-550). We do not believe that neophobia is a likely driver underlying our results, given that the outdoor laboratory has been used to collect data for many decades, including over a decade prior to the first field season in which data were sampled for our study (now highlighted in lines 692-694). In addition, previous studies at Bossou have determined that the outdoor laboratory is visited with comparable frequency to naturallyoccurring nut cracking sites, which makes any form of novelty bias unlikely (this information is now included in our methods, see lines 397-400, and also 687-689).

We agree that further information is required about foraging behaviours across the home range to understand changes in attendance at the outdoor laboratory, and have now provided more clarity on this within the limitations section of our discussion 542-550. In our discussion of individual survivability, we state clearly that we cannot make a conclusion about how changes in tool use influence survival with the available data, and assert that this would require data across the home range (see lines 627-638). We agree that future research is needed to assess whether changes in tool use would influence survivability, and also suggest that it may not be survival-relevant; instead changes in tool use with aging may simply be a litmus test for detecting more generalized senescence.

Generally speaking, there is a lack of consideration for temporal variation in ecological factors. As a control for these, Howard-Spink and colleagues have examined behavioral data for younger individuals from Bossou in the same years, to ostensibly show that patterns in older adults are different from patterns in younger adults, which is fair given the available data. Nonetheless, they seem to focus mostly on the start and end points and not patterns that occur in between. For example, there is a curious drop in attendance rate for all individuals in the 2008 season, the implications of which are not discussed by the authors.

As the reviewer points out, when examining the attendance rates of older individuals over sampled field seasons, we used the attendance rates of younger individuals as a control. However, we do not run this analysis using start and end points only. Attendance rates were included in our model across the full range of sample field seasons. However, as the key result here is an interaction term between age cohort (old) and the field season (scaled about the mean), we supplement this significant statistical result with a digestible comparison of attendance rates between the first and last field season, to give a general sense of effect size. We have clarified that all data were used in our model (see line 229, and also the legend for Table 2), and in this section we also provide all key model outputs and signpost where the full model output can be found in the supplementary materials.

As far as attendance, Howard-Spink and colleagues also discuss how this might be explained by changes in social standing in later life (i.e., chimpanzees move to the fringes of the social network and become less likely to visit gathering sites). This is not senescence in the sense of physiological and cognitive decline with older age. Instead, the reduced attendance due to changes in social standing seems rather to exacerbate signs of aging rather than be an indicator of it itself. The authors also mention a flu-like epidemic that caused the death of 5 individuals; the subsequent population decline and related changes in demography also warrant more discussion and characterization in the manuscript.

We have adapted this part of the discussion to make it clear that social aging is not necessarily equivalent to physiological and cognitive aging. We have also clarified in this section the changes in demography at Bossou during our study, which may have further impacted social behaviors (see lines 423-443).

Understandably, some of these issues cannot be evaluated or corrected with the presented dataset. Nonetheless, these undermine how certain and/or deterministic their conclusions can really be considered. Howard-Spink et al. have not strongly 'demonstrated' the validity of relationships between the variables of the study. If anything, their cursory observations provide us with methods to apply and hypotheses to test in future studies. It is likely that with higher-resolution datasets, the individual variability in age-related decline in tool-use abilities will be replicated. For now, this can be considered a starting point, which will hopefully inspire future attempts to research these questions.

We thank the reviewer for their comments. We have adapted our manuscript to highlight that we agree that it serves a starting point for answering these valuable questions; however, we do feel that we can contribute meaningful evidence that it is likely aging effects underlying the findings in our data (see responses above). We agree with the reviewer that further study is needed to understand these questions in more detail, and have tried to ensure that our conclusions are suitably tempered, and the recommendations for research are heavily encouraged to build on our findings.

Falótico, T., Valença, T., Verderane, M. & Fogaça, M. D. Stone tools differences across three capuchin monkey populations: food's physical properties, ecology, and culture. Sci. Rep. 12, 14365 (2022).

This has now been cited.

Forss, S. & Willems, E. The curious case of great ape curiosity and how it is shaped by sociality. Ethology 128, 552-563 (2022).

We do not cite this – see above.

Neufuss, J., Humle, T., Cremaschi, A. & Kivell, T. L. Nut-cracking behaviour in wild-born, rehabilitated bonobos (*Pan paniscus*): a comprehensive study of hand-preference, hand grips and efficiency. Am. J. Primatol. 79, e22589 (2016).

This has now been cited.

Proffitt, T., Reeves, J. S., Pacome, S. S. & Luncz, L. V. Identifying functional and regional differences in chimpanzee stone tool technology. R. Soc. Open Sci. 9, 220826 (2022).

This has now been cited.

Putt, S. S. The origins of stone tool reduction and the transition to knapping: An experimental approach. J. Archaeol. Sci.: Rep. 2, 51-60 (2015).

We do not cite this, as we instead cite studies which highlight chimpanzees’ ability to become more efficient in tool use with repeated practice (see above).

Tennie, C. & Call, J. Unmotivated subjects cannot provide interpretable data and tasks with sensitive learning periods require appropriately aged subjects: A Commentary on Koops et al. (2022) "Field experiments find no evidence that chimpanzee nut cracking can be independently innovated". ABC 10, 89-94 (2023).

We do not cite this – see above

**Reviewer #2 (Recommendations for the authors):**
Minor Comments:(1) Line 494: Citation #53 is listed twice.

This has been amended.

(2) Line 501: The term 'culturally-dependent' as used here is, at best, controversial, and at worst, misapplied. I would recommend replacing it with simply the term 'cultural'.

This has been changed to ‘cultural’.

Major Comments:For the Introduction, in the paragraph starting on Line 91, and the Discussion, starting on Line 369, I would recommend some simple re-structuring of the argumentation. As many in the Public Review, the changes in social standing according to age are not necessarily a case of senescence in the very sense of physiological or cognitive changes of the individual. This seems to have had an effect on attendance rates, which then could have been a driver of behavioral changes and even cognitive decline as ostensibly measured by the other variables. The social impact of aging should be mentioned in the Introduction (it is not currently) and the social and physiological/cognitive effects of aging should be separated in the Discussion. You can then discuss more clearly how the former via other behavioral changes can accelerate the latter (or not).

We take the point raised about social aging. Integrating information about social aging into the introduction was challenging without disrupting the flow of the paper; however, we have included these valuable points in the discussion (see lines 423-443). We now structure this section to clearly distinguish social aging, and discuss how, in tandem with changes in demography at Bossou, it may have influenced rates of attendance to the outdoor laboratory over the years. We do not go into detail about how social aging may interact with physiological or cognitive effects of aging, as we cannot support this with the available data, however we highlight at the end of this paragraph how all of these possible factors require further investigation.

For the present study, it will either be impossible or impractical to gather data on the yearly ecological conditions, contextualized dietary preferences, individual personalities, etc., so I would not ask that you do so. It is important, however, to temper some of the claims being made in the manuscript about what you have 'determined' about the nature of senescence in chimpanzees and to be more transparent about the limitations and potential confounds when interpreting the data. To avoid repetition, the key points can be found in the Public Review under 'Weaknesses'.

We appreciate the reviewer’s understanding of the limitations of our study. Some of these factors – such as individual personalities and dietary preferences – are addressed somewhat by our use of long-term data at the level of the individual, particularly in the analyses of efficiency, where we model individuals’ behaviors compared to those in earlier years offers an individuallybespoke control. However, there are other ecological variables of possible importance that we cannot evaluate. We now address several of these points raised by reviewers in the discussion, to ensure transparency of reporting (see limitations section of our discussion, and results to the comments provided by Reviewer 1, and our responses to points raised in the Public Review). We have also tempered some of the phrasing surrounding our conclusions, where we say that this is the first evidence that aging can impact chimpanzee tool use, we also highlight the need for an assortment of further studies.

Finally, the integration of the coula nut-cracking data is not well-executed as it stands. I would recommend that they collect and analyze equivalent behavioral data from the other years where coula nuts were provided. By examining only one season of coula nut-cracking, we cannot contextualize the data to past seasons; there is no sense in comparing one season of coula nut-cracking (i.e., in a sense of efficiency) to roughly contemporary seasons of palm-nut cracking due to, as you describe, differences in physical properties of the nuts. If you are not able to collect the additional data and carry out the requisite analysis, then I would recommend that the coula nut-related sections be removed from the manuscript, so that it does not detract from the logical flow of arguments and distract from the other data, which is more logically-attuned to your research questions.

We have removed this from the main manuscript. We have decided to include the information surrounding coula nut cracking in the supplementary materials, as this information is still relevant to the findings of our study, and may interest some readers. However, we have phrased this information to make it clear that further data is needed to compare coula nut cracking across years.

These criticisms do not subtract from the (potential) value or importance of the work for the field. This is, of course, an important contribution to an understudied topic. As such, I would gladly advocate for the manuscript, assuming the authors would reflect on the listed caveats and make changes in response to the 'Major Comments'.

We thank the reviewer for their comments.